# ROBUST REPRESENTATION CONSISTENCY MODEL VIA CONTRASTIVE DENOISING

**Jiachen Lei**[1,5], **Julius Berner**[2]*, **Jiongxiao Wang**[3], **Zhongzhu Chen**[4]
**Zhongjia Ba**[1], **Kui Ren**[1], **Jun Zhu**[5,6], **Anima Anandkumar**[7]
[1]Zhejiang University, [2]NVIDIA, [3]UW–Madison, [4]Amazon,
[5]Shengshu, [6]Tsinghua University, [7]Caltech

## ABSTRACT

Robustness is essential for deep neural networks, especially in security-sensitive applications. To this end, randomized smoothing provides theoretical guarantees for certifying robustness against adversarial perturbations. Recently, diffusion models have been successfully employed for randomized smoothing to purify noise-perturbed samples before making predictions with a standard classifier. While these methods excel at small perturbation radii, they struggle with larger perturbations and incur a significant computational overhead during inference compared to classical methods. To address this, we reformulate the generative modeling task along the diffusion trajectories in pixel space as a discriminative task in the latent space. Specifically, we use instance discrimination to achieve consistent representations along the trajectories by aligning temporally adjacent points. After fine-tuning based on the learned representations, our model enables implicit denoising-then-classification via a single prediction, substantially reducing inference costs. We conduct extensive experiments on various datasets and achieve state-of-the-art performance with minimal computation budget during inference. For example, our method outperforms the certified accuracy of diffusion-based methods on ImageNet across all perturbation radii by 5.3% on average, with up to 11.6% at larger radii, while reducing inference costs by 85× on average. Codes are available at: https://github.com/jiachenlei/rRCM.

## 1 INTRODUCTION

Deep neural networks (DNNs) have achieved unprecedented success in various visual applications. Yet, they are still vulnerable to small adversarial perturbations. This imposes a threat to the deployment of DNNs in real-world systems, in particular for security-critical scenarios, such as human face identification and autonomous driving. To counteract this issue, numerous efforts in terms of both empirical and certified defenses have been made to improve the robustness of DNNs against adversarial perturbations. While empirical defenses train DNNs to be robust to known adversarial examples (Mądry et al., 2017), they can be easily compromised by employing stronger or unknown perturbations. In contrast, to end the mouse-and-cat game of iterative improvements of attacks and defenses, *certified defenses* focus on developing strategies that provide certifiable and formal robustness guarantees. However, this also makes the design of such certified defenses much more challenging. Among certified defenses, *randomized smoothing with Gaussian noise* (Cohen

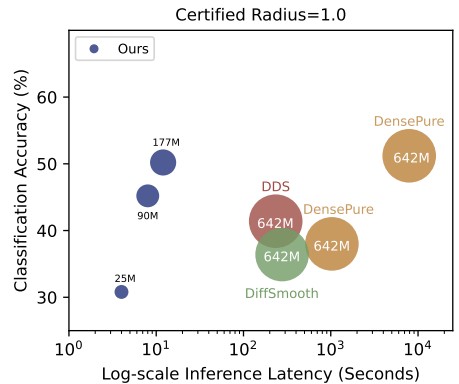

Figure 1: Performance vs. Inference Latency. Marker sizes correspond to relative model sizes.

---

*Work partially done at Caltech.

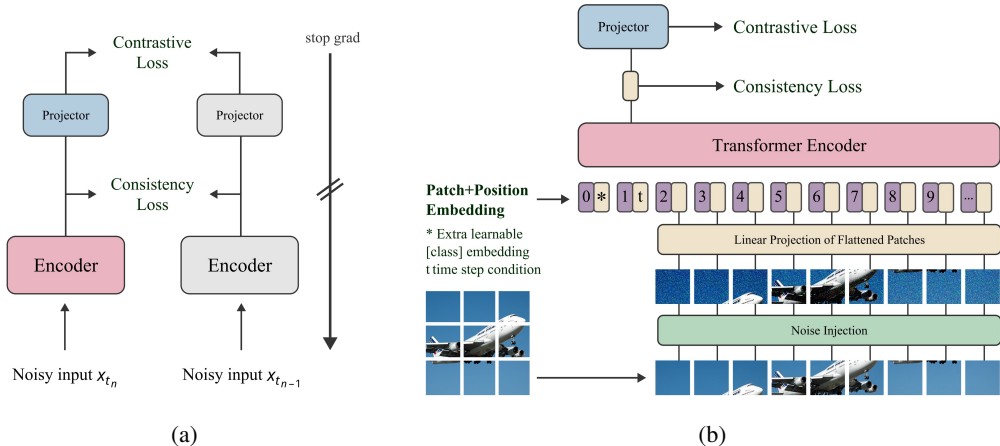

Figure 2: Illustration of our pre-training method and model forward pass. **(a)** Pre-training method. After pre-training, the projector is discarded, and the encoder is fine-tuned alongside a linear head using class labels. **(b)** Model forward pass. Noticeably, during certification, our model serves as the base classifier (as described in Section 2) and predicts the class label of each perturbed sample in a single forward pass.

et al., 2019) is currently considered the "gold standard", providing a scalable way of certifying model robustness against adversarial perturbations with bounded $\ell_2$-norm. To date, various randomized smoothing-based methods have been proposed (Jeong & Shin, 2020; Carlini et al., 2022). Among these works, diffusion model-based methods (Carlini et al., 2022) stand out with superior performance by integrating trained diffusion models into randomized smoothing. They first apply the denoising process of a diffusion model to remove Gaussian noise added to images. Then, using the purified samples, they predict the class label using a separate classifier. For brevity, we refer to these approaches as *diffusion-based methods* in the following discussions.

Despite the success, there exists a gap between achieving low latency and strong performance for diffusion-based methods. To maintain a competitive performance, they either increase the number of sampling steps (Xiao et al., 2022) and/or implement majority voting during class prediction (Xiao et al., 2022; Zhang et al., 2023), suffering from even higher computational demands during inference (e.g., as high as 52 minutes[1]). Furthermore, while leveraging the basic denoising property of diffusion models, previous approaches achieve consistent prediction across perturbed and clean samples through two independent models, resulting in a cumbersome prediction process and increased model maintenance overhead. We show that the framework of diffusion models itself already offers an effective solution in this regard: it establishes a unique connection between perturbed and clean samples along the trajectories of the probability flow (PF) of the denoising process. In this context, perturbed samples can be seen as points on the same trajectory of the denoising process but at different time steps, with the clean sample being the initial point[2]. This motivates our approach of directly optimizing for consistent semantics across noise-perturbed and clean samples on the same trajectory of the diffusion process, leading to a unified model that supports consistent one-step prediction. In contrast, classical methods (Cohen et al., 2019; Jeong & Shin, 2020; Salman et al., 2019a) train models directly on noisy samples, primarily relying on heuristic strategies. These approaches fail to thoroughly exploit the intrinsic relationships between noisy and clean images, limiting their potential to achieve higher levels of certified robustness.

**Our approach:** We close the gap of diffusion-based methods in terms of the tradeoff between performance and efficiency. In particular, we achieve performance that is better than classical randomized smoothing methods at a fraction of the cost of existing diffusion-based methods. This is made possible by directly optimizing model robustness based on structured connections between

---

[1]On a single A800 GPU, we report the time by certifying DensePure (Xiao et al., 2022) on a single image from ImageNet with $N$=10$k$ smoothing noises.

[2]As is common practice, we call the clean image on the reverse sampling trajectory the "initial point", as opposed to the one sampled from the Gaussian prior at the beginning of the reverse sampling process.

clean and perturbed samples. With the above analysis, we reformulate model robustness against noise perturbations as consistency between predictions of clean and perturbed samples. Specifically, our framework decomposes the training into two stages: pre-training and fine-tuning. During pre-training, the model learns to align representations across points along the deterministic trajectory from the Gaussian prior to the data distribution. To accomplish this, we reformulate the original generative image denoising task into a discriminative task in latent space and propose to align the representations of temporally adjacent points on the same trajectory via pair-wise instance discrimination. Based on the learned consistent representations, the model is then fine-tuned in a supervised manner to predict class labels given perturbed samples as input. This integrates denoising and classification into a single model and enables one-shot image classification, which is key to lowering the computation cost during inference. We term our model **Robust Representation Consistency Model (rRCM)**.

In our experiments, we demonstrate that our method improves the certified accuracy of Carlini et al. (2022) at all perturbation radii by 5.3% on average, with up to 11.6% at larger radius, while at the same time reducing the computation cost by $85\times$ on average during inference. In comparison with classical methods (Cohen et al., 2019; Jeong & Shin, 2020; Salman et al., 2019a; Zhai et al., 2020; Jeong et al., 2021), our rRCM model either matches or surpasses their performance, achieving an average improvement of 8.48% across all perturbation radii, and maintains a similar inference cost. Besides, we demonstrate that our method exhibits strong scalability w.r.t. the training budget, including model parameters and training batch size, on ImageNet (Deng et al., 2009). In particular, we empirically observe that the performance of our rRCM model has not yet plateaued, indicating that a larger training budget could lead to even higher certified robustness. These results underscore the advantages of leveraging established noise schedules from diffusion models to enhance model robustness and streamline the certification process, making our method more effective than previous approaches. Figure 1 illustrates the trade-off between performance and efficiency across different methods, including rRCM. In conclusion, our contributions are as follows:

1. **Structured noise schedule for robustness**. To the best of our knowledge, we are the first to exploit the advantages of the structured noise schedule of diffusion models in training robust classification models and provide a general direction for enhancing model robustness by drawing connections between noisy and clean samples.

2. **One-step denoising-then-classification**. Our method reformulates the denoising objective, a generative modeling task, in a discriminative manner. It supports one-step denoising-then-classification, lowering computational demands and maintenance overhead. Besides, it offers remarkable representation consistency in the sense that our model is capable of generating meaningful representations by mapping random noise to the manifold of the clean data in latent space.

3. **Bridging efficiency-performance trade-offs**. Our method bridges the gap in achieving low latency and superior performance for diffusion-based randomized smoothing methods. We perform extensive experiments across various datasets, demonstrating that rRCM achieves state-of-the-art performance compared to existing methods.

4. **Strong scalability**. Our training framework also exhibits strong scalability w.r.t. enhancing model robustness on large-scale datasets like ImageNet.

## 2 PRELIMINARIES

**Diffusion Models** (Ho et al., 2020; Song et al., 2020) aim to approximate the underlying data distribution $p(\boldsymbol{x}_0)$, given training data $\boldsymbol{x}_0 \sim p(\boldsymbol{x}_0)$. They are composed of a forward and reverse process. Following the definition of Karras et al. (2022), the forward process of a diffusion model is given by the stochastic differential equation (SDE) $d\boldsymbol{x}_t = \sqrt{2t}d\boldsymbol{w}_t$, where $\boldsymbol{w}_t$ is the standard Wiener process and $t \in [0, T]$ (we use $T = 80$). Let $p_t(\boldsymbol{x})$ denotes the data distribution of the solution $\boldsymbol{x}_t$ to the forward SDE at time $t$, where $p_T(\boldsymbol{x}) \approx \mathcal{N}(0, T^2\boldsymbol{I})$. Correspondingly, the reverse process is given by the reverse-time SDE $d\boldsymbol{x}_t = -2t\nabla_{\boldsymbol{x}} \log p_t(\boldsymbol{x})dt + \sqrt{2t}d\bar{\boldsymbol{w}}_t$, starting at $t = T$. Here, $\bar{\boldsymbol{w}}_t$ is a standard Wiener process that runs backward in time and $\boldsymbol{x}_T \sim p_T(\boldsymbol{x})$. For the given reverse-time SDE, there exists a corresponding deterministic reverse-time process that shares the same marginal probability densities $\{p_t(\boldsymbol{x})\}_{t \in [0,T]}$ as the SDE:

$$d\boldsymbol{x}_t = -t\nabla_{\boldsymbol{x}} \log p_t(\boldsymbol{x})dt \tag{1}$$

The ordinary differential equation (ODE) given above is referred to as *Probability Flow ODE* (PF ODE) (Song et al., 2020). Given the score function $\nabla_{\boldsymbol{x}} \log p_t(\boldsymbol{x})$, one can generate "clean" samples from $p(\boldsymbol{x}_0)$ by sampling from the prior $\mathcal{N}(0, T^2 \boldsymbol{I})$ and following the deterministic trajectories given by the above PF ODE.

As is common practice, we consider the discretized versions of the above equations. Using time steps $\{t_n\}_{n=0}^N$, we divide the time horizon into $N$ non-overlapping intervals. The endpoints, $t_0$ and $t_N$, are chosen such that $\boldsymbol{x}_{t_0}$ can be seen as approximate sample from $p(\boldsymbol{x}_0)$ and $t_N = T$. We denote points along the PF ODE sampling trajectory as $\{\boldsymbol{x}_{t_n}\}_{n=0}^N$. Subsequently, the PF ODE in (1) can be discretized as

$$\boldsymbol{x}_{t_{n-1}} = \boldsymbol{x}_{t_n} - t_n(t_{n-1} - t_n)\nabla_{\boldsymbol{x}} \log p_{t_n}(\boldsymbol{x})|_{\boldsymbol{x}=\boldsymbol{x}_{t_n}}. \tag{2}$$

Given a clean sample $\boldsymbol{x}_0$, one can also directly sample $\boldsymbol{x}_{t_n}$ using

$$\boldsymbol{x}_{t_n} = \boldsymbol{x}_0 + t_n \boldsymbol{\epsilon} \quad \text{with} \quad \boldsymbol{\epsilon} \sim \mathcal{N}(0, \boldsymbol{I}). \tag{3}$$

**Randomized Smoothing** (Cohen et al., 2019) is a technique to certify the robustness of arbitrary classifiers against adversarial perturbations under the $\ell_2$-norm. It leverages a base classifier's robustness against random noise and builds a new classifier robust to adversarial perturbations, providing theoretical guarantees for the robustness of this new classifier. Given an input sample $\boldsymbol{x}$ and the base classifier $f$ with classes $\mathcal{Y}$, randomized smoothing considers a smoothed version of $f$ defined as

$$F(\boldsymbol{x}) = \arg\max_{c \in \mathcal{Y}} \ \mathbb{P}_{\boldsymbol{\epsilon} \sim \mathcal{N}(0,\boldsymbol{I})}\big[f(\boldsymbol{x} + \sigma \cdot \boldsymbol{\epsilon}) = c\big], \tag{4}$$

where the noise level $\sigma$ is a hyper-parameter of the smoothed classifier. In our following discussion, we term $F$ as the *hard model* and $f$ as the *soft model*. Suppose $f$ classifies samples from $\mathcal{N}(\boldsymbol{x}, \sigma^2 \boldsymbol{I})$, the predicted probability of the most probable class is $p_{c_A}$, and the runner up probability is $p_{c_B}$. Then, the robustness radius lower bound $r$ of the hard model $F$ around $\boldsymbol{x}$ is given by

$$r = \frac{\sigma}{2}\bigg(\Phi^{-1}\big(p_{c_A}\big) - \Phi^{-1}\big(p_{c_B}\big)\bigg), \tag{5}$$

where $\Phi^{-1}$ is the inverse cumulative distribution function (CDF) of a standard Gaussian distribution. To achieve strong robustness under noise perturbations, one should maintain consistent predictions across clean and noisy samples. In practice, we adopt the common setting of classification models and parameterize the soft model as a function that outputs logits, which then pass through a softmax operation to obtain discrete probabilities of $\boldsymbol{x}$ belonging to each of the classes. Moreover, to approximate the probability in (4), we use a large number (typically 10k or 100k in practice) samples of $\boldsymbol{\epsilon}$, so-called *smoothing noises*.

## 3 METHOD

### 3.1 OVERVIEW

A robust model shall give consistent predictions across clean and perturbed samples. To achieve this, we first draw connections between clean and perturbed samples leveraging the PF ODE used in diffusion models. For a given initial condition, the PF ODE ensures that trajectories remain distinct and do not cross each other. This indicates that any point uniquely belongs to a single sampling trajectory. In this context, points on the same trajectory can be interpreted as data of the same latent representation, defined by the initial clean data point $\boldsymbol{x}_0$. By formulating the training objective as grouping points on the same trajectory (in latent space), we can align points at higher noise levels with those from earlier time steps with lower noise levels, ultimately reaching the consistency goal.

We achieve this goal in a two-step process: pre-training and fine-tuning. During pre-training, we treat both clean and perturbed samples as points along the same deterministic PF ODE sampling trajectory of the diffusion model defined in Section 2. We align representations between temporally adjacent points that are sampled along the trajectory via pair-wise instance discrimination. Specifically, we attract temporally adjacent points on the same trajectory, while repelling those from different trajectories, leading to consistent representations among perturbed and clean samples. Afterwards, drawing upon the acquired consistent representations, we fine-tune the model via supervised training with class labels and additionally enforce consistent predictions on perturbed samples of the same noise magnitude. This transforms the alignment task from sample-to-sample alignment

during pre-training to sample-to-class-label alignment and further partitions the trajectories based on their respective classes.

Our approach reframes the image denoising task as a discriminative task in the latent space, effectively learning **denoising by discrimination**. This unifies the two independent modules into a single model and enables robust one-shot image classification. Next, we formalize the aforementioned idea and provide a detailed description of our training methodology.

### 3.2 PROBLEM FORMULATION

Given points along the PF ODE sampling trajectory of the diffusion model, our goal is to align the logits produced by the soft model $f_\phi$, with $\phi$ as model parameters. This can be formulated as

$$\arg\max_\phi \big(\hat{f}_\phi(\boldsymbol{x}_{t_n}) \cdot \hat{f}_\phi(\boldsymbol{x}_{t_{n-1}})\big) \quad \text{with} \quad \hat{f}_\phi = \frac{f_\phi}{||f_\phi||}, \quad \forall n \in \{1, ..., N\}. \tag{6}$$

Her $\{\boldsymbol{x}_{t_n}\}_{n=0}^N$ are obtained as in (2). The alignment objective in (6) maximizes the cosine similarity between paired samples $(\boldsymbol{x}_{t_n}, \boldsymbol{x}_{t_{n-1}})$, similar to the goal of contrastive learning methods (Chen et al., 2020; He et al., 2020; Chen & He, 2021; Chen et al., 2021), which attracts semantically similar views (*positive pairs*) of a sample in latent space while repelling dissimilar ones (*negative pairs*). In our case, the positive pairs are temporally adjacent points along the same PF ODE sampling trajectory, while points from different trajectories serve as negative pairs.

Considering the similar underlying rationale, we decompose the training into two stages (pre-training and fine-tuning), and we parameterize the soft model $f_\phi$ as $f_{\phi=\{\omega,\theta\}} = h_w \circ g_\theta$, where $g_\theta$ is a neural network and $h_w$ is a linear layer. During pre-training, we train $g_\theta$ to align points along the same PF ODE sampling trajectory in latent space. Then, we fine-tune $g_\theta$ together with the linear head $h_w$ using class labels, ultimately achieving consistent class prediction on perturbed images. We will discuss details of our pre-training and fine-tuning method next.

### 3.3 PRE-TRAINING

Given a sequence of i.i.d. samples[3] $\mathcal{X} = \{\boldsymbol{x}_0^i\}_{i=1}^B$ drawn from $p(\boldsymbol{x}_0)$, we aim to reformulate the alignment objective using loss functions similar to the infoNCE loss (Oord et al., 2018), i.e.,

$$L(\mathcal{A}, g_\theta, \mu) = \mathbb{E}_\mathcal{A}\left[\sum_{i=1}^B \left(-\log \frac{G_\theta(\boldsymbol{a}_1^i, \boldsymbol{a}_2^i; g_\theta, \mu)}{\sum_{j=1}^B G_\theta(\boldsymbol{a}_1^i, \boldsymbol{a}_2^j; g_\theta, \mu)}\right)\right], \tag{7}$$

where

$$G_\theta(\boldsymbol{u}, \boldsymbol{v}; g_\theta, \mu) = \exp\left(\frac{\hat{g}_\theta(\boldsymbol{u}) \cdot \hat{g}_{\theta^-}(\boldsymbol{v})}{\tau}\right) \quad \text{with} \quad \hat{g}_\theta(\boldsymbol{u}) = \frac{g_\theta(\boldsymbol{u})}{||g_\theta(\boldsymbol{u})||}. \tag{8}$$

In the above, $\theta^-$ is an exponential moving average (EMA) of $\theta$ with EMA update rate $\mu$, and $\tau$ is a hyper-parameter (that we set to 0.2 in our experiments). Moreover, $\mathcal{A} = \{(\boldsymbol{a}_1^i, \boldsymbol{a}_2^i)\}_{i=1}^B$ is a sequence of samples, where $\boldsymbol{a}_1^i$ and $\boldsymbol{a}_2^i$ are considered a positive sample pair defining two related yet different views of the $i$-th sample $\boldsymbol{x}_0^i$ while $\{\boldsymbol{a}_2^j\}_{j \neq i}$ are treated as negative samples to the $i$-th sample. Overall, our alignment objective is then given by

$$\arg\min_\theta \; L(\mathcal{X}, g_\theta, \mu_1) + L(\mathcal{Z}, p_\nu \circ g_\theta, \mu_2). \tag{9}$$

The differences between the two terms lie in three aspects: the construction of positive and negative pairs, model for computing the loss, and EMA update rate $\mu$. We call the first term in (9) the *consistency loss* and the second one the *contrastive loss*. Next, we will discuss how to construct the positive and negative samples for each loss.

In the contrastive loss, i.e., the second term in (9), $\mathcal{Z}$ denotes a sequence of augmented samples created following the convention in contrastive learning literature (Chen et al., 2020; 2021). Specifically, we construct each positive pair by applying different data augmentations to the clean data $\boldsymbol{x}_0^i$

---

[3]We use subscripts to distinguish clean samples $\boldsymbol{x}_0$ from noisy samples $\boldsymbol{x}_{t_n}$ and superscripts to denote different clean samples $\boldsymbol{x}^i$.

and consider other augmented samples within the same batch as negative samples to the $i$-th sample. For the consistency loss, i.e., the first term in (9), we define $\mathcal{X} = \{(\boldsymbol{x}_{t_n}^i, \boldsymbol{x}_{t_{n-1}}^i)\}_{i=1}^B$, where we uniformly sample a unique $n$ in $\{1, \ldots, N\}$ for each batch of samples. We consider $\boldsymbol{x}_{t_n}^i$ and $\boldsymbol{x}_{t_{n-1}}^i$ as a positive sample pair, representing temporally adjacent points from the same PF ODE trajectory that share the same clean image $\boldsymbol{x}_0^i$. Moreover, $\{\boldsymbol{x}_{t_{n-1}}^j\}_{j \neq i}$ are treated as negative samples to the $i$-th sample $\boldsymbol{x}_{t_n}^i$. They are constructed with other samples within the same batch and are temporally adjacent to $\boldsymbol{x}_{t_n}^i$ yet from different PF ODE trajectories.

To construct a positive pair $(\boldsymbol{x}_{t_n}^i, \boldsymbol{x}_{t_{n-1}}^i)$ with clean data $\boldsymbol{x}_0^i$, we first sample $\boldsymbol{x}_{t_n}^i$ following the discrete forward process in (3). Then, we could compute $\boldsymbol{x}_{t_{n-1}}^i$ by (2). However, the score $\nabla_{\boldsymbol{x}} \log p_{t_n}(\boldsymbol{x})$ is unknown. To address this, one could employ a pre-trained score model and perform a single-step denoising given $\boldsymbol{x}_{t_n}$. Alternatively, the score can also be expressed via Tweedie's formula (Efron, 2011), i.e.,

$$\nabla_{\boldsymbol{x}} \log p_{t_n}(\boldsymbol{x}) = -\mathbb{E}\left[\frac{\boldsymbol{x}_{t_n} - \boldsymbol{x}_0}{t_n^2} \Big| \boldsymbol{x}_{t_n}\right]. \tag{10}$$

Following (Song et al., 2023), we can then use a Monte Carlo estimate of the expectation and approximate $\boldsymbol{x}_{t_{n-1}}^i$ as

$$\boldsymbol{x}_{t_{n-1}}^i = \boldsymbol{x}_{t_n}^i + (t_{n-1} - t_n)\boldsymbol{\epsilon}. \tag{11}$$

Notably, each positive pair $(\boldsymbol{x}_{t_n}^i, \boldsymbol{x}_{t_{n-1}}^i)$ shares the same Gaussian noise $\boldsymbol{\epsilon}$. We adopt this method in our work, leaving further exploration of pre-trained score models to future research.

The contrastive loss in (9) (second term) is incorporated to enhance the model's semantic discrimination capabilities, enabling it to better distinguish points on different trajectories, particularly at early time steps, and ultimately improving certified robustness. Otherwise, the model tends to rely on trivial representations, which leads to training difficulties. To this end, we additionally include an extra projector head $p_\nu$, a 3-layer MLP, alongside the encoder $g_\theta$ during pre-training. The projector head acts as an information bottleneck that focuses on learning augmentation-invariant representations (Chen et al., 2020), and is removed later during fine-tuning. In early experiments, we observe that computing both losses on the output of the projector head leads to training instabilities. Therefore, we do not employ the projector when computing consistency loss.

We refer to Figure 2a for an overview of our pre-training method. We illustrate details of our model forward pass in Figure 2b. In particular, the input to the model includes a time embedding, a learnable class token, and noisy image tokens. The time embedding is included to provide the model with awareness of the noise magnitude added to the input samples, following the practice established in diffusion models (Song et al., 2020). When computing the loss, we select the output token corresponding to the learnable class token. As mentioned earlier, the consistency loss is calculated using the token from the model's output, while the contrastive loss is computed using the token from the projector's output.

For brevity, we name $g_\theta$ as *online model* and $g_{\theta^-}$ as *target model*. For the two loss terms, we use two separate target models, each parameter-

**Algorithm 1** rRCM Pre-training Pseudocode

```
# g: online model
# g_ema: target model
# proj: the projector head
# z1 and z2: two augmented views of x0
# t1 and t2: two adjacent time steps
# epsilon: Gassian noise sampled
# from N(0, I)
for z1, z2, x0, tn in data_loader:
    eps = randn_like(x0)
    xt1 = x0 + t1*epsilon
    xt2 = x0 + t2*epsilon
    f1 = proj(g(z1, t0))
    f2 = proj(g_ema(z2, t0))
    p1 = g(xt1, t1)
    p2 = g(xt2, t2).detach()
    loss = consistency_loss(p1, p2)
    loss += contrastive_loss(f1, f2)
    loss.backward()
    update(g_ema)
```

ized by a distinct $\theta^-$ and updated with different EMA rates. Specifically, we set $\mu_1$ and $\mu_2$ in (9) to 0 and 0.99 respectively. To avoid maintaining two sets of frozen parameters, we simplify the process for the consistency loss by using the online model directly and stopping gradient back propagation from the resulting model output. The pseudocode for the pre-training procedure is provided in Algorithm 1. While conceptually similar to contrastive learning, the pretraining of rRCM significantly differs from previous methods. To demonstrate this, we conduct further experiments and compare the effectiveness of our method with MoCo-v3 (Chen et al., 2021), additionally equipped

with Gaussian noise augmentation, in Appendix F. We also present detailed comparisons with contrastive learning methods and consistency model Song et al. (2023) in Section 5.

### 3.4 FINE-TUNING

As described in Section 3.1, during fine-tuning, we map each perturbed sample to its ground-truth label while enforcing consistent predictions among among samples generated via the forward SDE, given the same clean image at the same time step $t$. In our work, we adopt the diffusion model proposed in EDM (Karras et al., 2022) and the time step $t$ is interchangeable with the noise level $\sigma$ in (4), as can be seen in (3). For randomized smoothing, $\sigma$ typically takes values in $\{0.25, 0.5, 1.0\}$. In our experiments, starting with the same pre-trained weights $\theta$, we fine-tune the model $f_{\phi=\{w,\theta\}}$ independently for each noise level. Specifically, for a given noise level $\sigma$, we fine-tune the model using the following training objective (Jeong & Shin, 2020)

$$\arg\min_{\phi} \mathbb{E}_{\boldsymbol{x}_{\sigma},\boldsymbol{x}'_{\sigma}} \Big[ -p(c)\log(p_{\phi}(\boldsymbol{x}_{\sigma})) - \eta_1 \cdot p_{\phi}(\boldsymbol{x}_{\sigma})\log p_{\phi}(\boldsymbol{x}'_{\sigma}) - \eta_2 \cdot p_{\phi}(\boldsymbol{x}_{\sigma})\log p_{\phi}(\boldsymbol{x}_{\sigma}) \Big]. \quad (12)$$

Here, $p_{\phi}(\boldsymbol{x}_{\sigma}) = \text{softmax}(f_{\phi}(\boldsymbol{x}_{\sigma}))$ and $\boldsymbol{x}_{\sigma} = \boldsymbol{x} + \sigma\boldsymbol{\epsilon}$ and $\boldsymbol{x}'_{\sigma} = \boldsymbol{x} + \sigma\boldsymbol{\epsilon}'$ are two noisy versions of $\boldsymbol{x} \sim p(\boldsymbol{x}_0)$, where $\epsilon, \epsilon' \sim \mathcal{N}(0, \boldsymbol{I})$. The variable $c$ denotes the class label of the sample $\boldsymbol{x}$ and $\eta_1, \eta_2$ are hyper-parameters. In the above, the first two terms represent the cross-entropy loss, which aligns the model's predictions with the ground-truth label and enforces consistency between predictions for the two perturbed versions of the same input. The third term computes the entropy of the model's predictions, acting as a regularization mechanism. This regularization encourages the model to make confident class predictions, contributing to achieving a larger robustness radius.

In early experiments, we observed that training a ViT model from scratch with this objective proved challenging. Upon further analysis, we speculate that the model struggles to simultaneously learn meaningful representations for class predictions while ensuring consistent predictions for noisy samples derived from the same clean image. However, after pre-training with our objective in (9), the model converges smoothly. We attribute this improvement to the similar representations among perturbed samples acquired during pre-training. We defer detailed explanation of the underlying rationale of our fine-tuning method to Appendix E.

## 4 EXPERIMENTS

### 4.1 EXPERIMENT SETTINGS

In this section, we evaluate our rRCM model on two datasets: ImageNet (Deng et al., 2009) and CIFAR10 (Krizhevsky et al., 2009). First, we demonstrate the efficiency and effectiveness of rRCM in comparison with existing baseline methods. Second, we study the scalability of our method in the aspects of model size and training batch size. We defer training details and the ablation studies on hyper-parameters of our method to the Appendix.

**Model.** We employ three different models in our experiments, namely, rRCM-S, rRCM-B , and rRCM-B-Deep, with an increasing number of parameters. All models follow the Vision Transformer (ViT) architecture (Dosovitskiy et al., 2020). Unless otherwise specified, we conduct experiments on ImageNet with rRCM-B and rRCM-B-Deep model, and conduct experiments on CIFAR10 with rRCM-B model. Further details on our model architectures can be found in the Appendix.

**Certification.** We follow the settings of Carlini et al. (2022). Specifically, on both ImageNet and CIFAR10, we certify a subset that contains 500 images from their test set with confidence 99.9%. We certify each sample at three different noise levels $\sigma \in \{0.25, 0.5, 1.0\}$, and report the certified accuracy under different perturbation radii $r$. We report certified accuracies of rRCM models utilizing both $10,000$ and $100,000$ smoothing noises on ImageNet, and $100,000$ smoothing noises on CIFAR10. We compare our models with a series of baseline methods. Both on ImageNet and CIFAR10, the certified accuracy of classical methods (Salman et al., 2020; Jeong & Shin, 2020; Salman et al., 2019a; Horváth et al., 2021; Zhai et al., 2020; Jeong et al., 2021) is reported utilizing $100,000$ smoothing noises. We measure the inference latency of all methods on a single A800 GPU.

---

[4]We attribute the reduced time expense compared to classical methods to the use of advanced deep learning code toolkits, such as *xFormers* (https://github.com/facebookresearch/xformers)

Table 1: Results on ImageNet. [1]We report the latency of classical randomized smoothing methods based on the number we obtained on Gaussian (Carlini et al., 2022). [2]We report the number from DiffSmooth (Zhang et al., 2023). [‡]Evaluated with 10,000 smoothing noises. Following the notations in (Xiao et al., 2022; Zhang et al., 2023), we denote the total number of model predictions utilized in majority voting with $K$ and $m$ respectively.

| Method | Latency[1] | Certified Accuracy at $r$ (%) | | | | | |
|---|---|---|---|---|---|---|---|
| | | 0.0 | 0.5 | 1.0 | 1.5 | 2.0 | 2.5 |
| Gaussian (Salman et al., 2019a) | 1min 20s | 67.0 | 49.0 | 37.0 | 29.0 | 19.0 | 15.0 |
| Consistency (Jeong & Shin, 2020) | 1min 20s | 55.0 | 50.0 | 44.0 | 34.0 | 24.0 | 21.0 |
| SmoothAdv (Salman et al., 2019a) | 1min 20s | 67.0 | 56.0 | 43.0 | 37.0 | 27.0 | **25.0** |
| Boosting (Horváth et al., 2021) | 4min | 65.6 | 57.0 | 44.6 | 38.4 | 28.6 | 24.6 |
| MACER (Zhai et al., 2020) | 1min 20s | 68.0 | 57.0 | 43.0 | 31.0 | 25.0 | 18.0 |
| SmoothMix (Jeong et al., 2021)[2] | 1min 20s | 55.0 | 50.0 | 43.0 | 38.0 | 26.0 | 24.0 |
| Denoised (Salman et al., 2020) | - | 60.0 | 33.0 | 14.0 | 6.0 | - | - |
| DDS[‡] (Carlini et al., 2022) | 3min 52s | 76.2 | 61.0 | 41.4 | 28.0 | 21.2 | 17.2 |
| DensePure[‡] (Xiao et al., 2022) K=1 | 17min 8s | 76.6 | 57.0 | 38.0 | 22.2 | 17.0 | 13.2 |
| K=5 | 52min 20s | 77.8 | 64.6 | 38.4 | 23.0 | 18.4 | 14.0 |
| DiffSmooth[‡] (Zhang et al., 2023) m = 5 | 4min 41s | 70.1 | 59.7 | 34.7 | 24.8 | 18.0 | 13.8 |
| m = 10 | 5min 10s | 70.0 | 61.4 | 36.0 | 26.4 | 20.8 | 18.0 |
| m = 15 | 5min 35s | 69.8 | 62.2 | 36.4 | 28.2 | 21.6 | 19.2 |
| **rRCM-B[‡]** | 6s | 76.6 | 62.6 | 45.2 | 33.8 | 27.0 | 22.0 |
| **rRCM-B** | 53s[4] | 76.8 | 63.0 | 45.6 | 34.8 | 28.0 | 22.6 |
| **rRCM-B-Deep** | 1min 41s | **77.4** | **64.0** | **51.2** | **40.0** | **32.6** | **25.0** |

## 4.2 MAIN RESULTS

On both datasets, we report both the time cost (latency) of certifying one sample and the classification accuracy under various perturbation radii. The results of our rRCM models on ImageNet and CIFAR10 are shown in Table 1 and Table 2, respectively. As demonstrated, we achieve superior performance over current diffusion-based randomized smoothing methods (Carlini et al., 2022; Xiao et al., 2022; Zhang et al., 2023) especially at large perturbation radii, while significantly reducing the computational cost, which is on par with other classical methods (Salman et al., 2019a; Jeong & Shin, 2020; Salman et al., 2020; Horváth et al., 2021; Zhai et al., 2020; Jeong et al., 2021).

**Performance on ImageNet.** As shown in Table 1, in comparison with both classical and diffusion-based methods, our rRCM-B model yields superior performance while maintaining an inference cost (53 seconds) slightly lower than that of classical methods (1 minutes and 20 seconds). Performance can be further improved by using a deeper model, rRCM-B-Deep, which ultimately reaches state-of-the-art results. This demonstrates the promising scalability of our approach, as detailed in Section 4.3.

Subsequently, we also conduct fine-grained experiments to demonstrate the unwilling computation trade-off of DensePure (Xiao et al., 2022) and DiffSmooth (Zhang et al., 2023) in order to achieve competitive results to classical methods, in particular at large perturbation radii. In specific, we re-implement DDS (Carlini et al., 2022), DensePure, and DiffSmooth under the recommended settings in respective works. For DDS and DensePure, we use a ViT-based classifier that has the same amount of parameters as our rRCM-B model and achieves 81.35% accuracy on the ImageNet validation set. For DiffSmooth, we follow the settings in Zhang et al. (2023) and use the same base classifier as DDS and DensePure but instead fine-tuned respectively with samples augmented with Gaussian noise at various noise levels $\sigma \in \{0.25, 0.5, 1.0\}$. We report their certified accuracies utilizing $10,000$ smoothing noises under different $\ell_2$ radii.

As anticipated, when the computation budget is limited and only a small number of majority voting is adopted during class prediction, both DensePure and DiffSmooth exhibit poorer performance than that of DDS. Noticeably, while adopting more denoising steps (b=5) during purification process, DensePure yields worse performance than DDS when no majority voting is applied during

class prediction. As we increase the majority voting number, the performance of both methods gradually increase at different pace. Though finally surpassing DDS, their computation overhead increases tremendously, a phenomenon especially observed on results of DensePure, which requires 52 minutes and 20s for certifying a single sample.

**Performance on CIFAR10.** As shown in Table 2, we reach superior certified classification accuracy, pushing the certified accuracy of DDS (Carlini et al., 2022) up at most by 6.4% ($r = 0.5$). Besides, our rRCM-B model either surpasses or is highly competitive to other high-performing methods, including SmoothAdv (Salman et al., 2019a), Boosting (Horváth et al., 2021), and MACER (Zhai et al., 2020). Our rRCM-B model is outperformed at $r = 0.75$ by Boosting (Horváth et al., 2021), a method that ensembles 10 different classifiers. Yet, we still surpass diffusion-based methods at all perturbation radii.

Table 2: Results on CIFAR10. [1]We report the latency of standard randomized smoothing methods based on the results we obtained on Gaussian (Carlini et al., 2022).

| Method | Latency[1] | Certified Accuracy at $r$ (%) | | | | |
|---|---|---|---|---|---|---|
| | | 0.0 | 0.25 | 0.5 | 0.75 | 1.0 |
| Gaussian (Cohen et al., 2019) | 4s | 83.0 | 61.0 | 43.0 | 32.0 | 22.0 |
| Consistency (Jeong & Shin, 2020) | 4s | 77.8 | 68.8 | 58.1 | 48.5 | 37.8 |
| SmoothAdv (Salman et al., 2019a) | 4s | 82.0 | 68.0 | 54.0 | 41.0 | 32.0 |
| Boosting (Horváth et al., 2021) | 40s | 83.4 | 70.6 | 60.4 | **52.4** | 38.8 |
| MACER (Zhai et al., 2020) | 4s | 81.0 | 71.0 | 59.0 | 46.0 | 38.0 |
| SmoothMix (Jeong et al., 2021) | 4s | 77.1 | 67.9 | 57.9 | 47.7 | 37.2 |
| DDS Carlini et al. (2022) | 52s | 79.8 | 69.9 | 55.0 | 47.6 | 37.4 |
| DiffSmooth Zhang et al. (2023) | 3min 34s | 78.2 | 67.2 | 59.2 | 47.0 | 37.4 |
| **rRCM-B** | 16s | **83.6** | **73.4** | **61.4** | 48.0 | **39.2** |

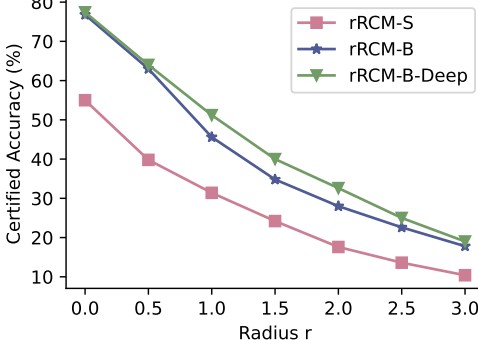

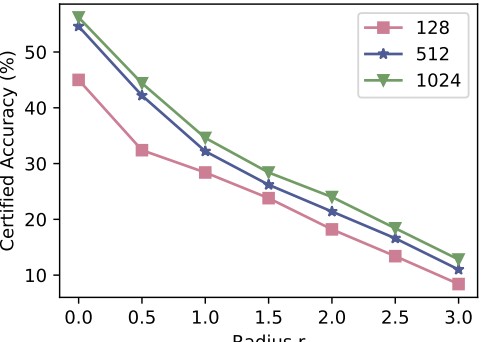

Figure 3: Scaling up model size on ImageNet improves performance.

Figure 4: Increasing training batch sizes on ImageNet improves performance.

## 4.3 SCALABILITY

We now explore the scalability of our method by pre-training models with varying model parameters and batch sizes on the ImageNet dataset. Following our experiment settings in Section 4.2, we additionally train a rRCM-S model and compare the certified accuracy of rRCM-S, rRCM-B, and rRCM-B-Deep. Additionally, utilizing rRCM-B, we investigate the impact of training batch size on model performance. The results, presented in Figures 3 and 4, highlight the excellent scalability of our method. With increased computational resources, we anticipate further performance improvements, which we leave for future work.

## 5 RELATED WORK

**Certified Robustness**. Deep neural networks (DNNs) are susceptible to adversarial examples (Goodfellow et al., 2014), prompting the development of various defense techniques, includ-

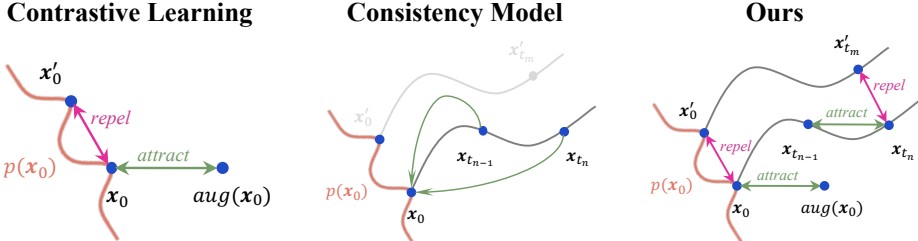

Figure 5: Comparisons of contrastive learning and consistency model training with our method. The gray lines denote the PF ODE trajectories. $x_0$ and $x_0'$ are two different clean samples that act as the initial point on respective PF ODE trajectory.

ing empirical defense and certified robustness. While empirical defense methods (Mądry et al., 2017; Samangouei, 2018; Zhang et al., 2019) can be easily compromised utilizing stronger adaptive attacks, certified robustness aims at providing a theoretical guarantee for the lower bound of model prediction accuracy under constrained perturbations. In certified robustness, a series of efforts (Raghunathan et al., 2018a;b; Salman et al., 2019b; Zhang et al., 2018)have been devoted to provide a robustness certification of DNNs. However, randomized smoothing (Lecuyer et al., 2019; Cohen et al., 2019) attract most attention due to its superior scalability. It supports non-trivial certification on large-scale dataset such as ImageNet and is applicable to any model architectures. On top of this, numerous works (Jeong & Shin, 2020; Salman et al., 2019a; Horváth et al., 2021; Zhai et al., 2020; Jeong et al., 2021; Li et al., 2024; Jeong & Shin, 2024) have been proposed to further enhance model's robustness. To the best of our knowledge, we are the first to utilize a structured noise schedule to train randomized smoothing based model for enhanced adversarial robustness.

**Teacher-Student training paradigm** is widely adopted in various domains, including representation learning and generative modeling. Contrastive Learning(Chen et al., 2020; Chen & He, 2021; He et al., 2020) aims at capturing meaningful visual representation by encouraging the model to output similar representations for samples of similar semantics. Meanwhile, as a member of score-based generative models (Ho et al., 2020; Song et al., 2020; Karras et al., 2022), consistency model (Song et al., 2023), a variant of diffusion models, employs a two-branch network to approximate the analytical solution of the PF ODE at initial point, resulting in consistent image predictions given any points on the same PF ODE sampling trajectory. Here, the clean image serves as a static boundary condition, preventing the model from learning trivial solutions. In comparison, rather than learning superior visual representations or achieving consistent image prediction, we aim at strong model robustness against adversarial perturbations. We learn consistent representations across points on the PF ODE trajectory by discriminating whether given point pairs are from the same PF ODE sampling trajectory. Besides, The initial point we utilize is low-dimensional representation dynamically learned during the training process. Noticeably, our rRCM model operates directly on image inputs, differing significantly from two-stage generative methods like LCM (Luo et al., 2023) which trains a consistency model in the latent space of a pre-trained VAE (Kingma, 2013). We present comparisons of contrastive learning, consistency model with rRCM in Figure 5.

## 6 CONCLUSION

In this work, we introduce the Robust Representation Consistency Model (rRCM), a novel approach to enhancing model robustness against adversarial perturbations through contrastive denoising in latent space. By reformulating the generative modeling process as a discriminative task, rRCM leverages a structured noise schedule to align representations of noisy and clean samples, allowing for one-step denoising and classification. This integration enables substantial reductions in inference costs, outperforming existing diffusion-based smoothing methods by a notable margin, particularly at higher perturbation radii. Our evaluations on ImageNet and CIFAR-10 confirm that rRCM achieves state-of-the-art performance with significantly improved efficiency, bridging the gap in the trade-off between robustness and latency. The proposed framework not only offers a promising approach to certified robustness but also establishes a foundation for future applications in representation learning and image generation. We leave further exploration of these applications for our future work.

ACKNOWLEDGMENTS

J.B. acknowledges support from the Wally Baer and Jeri Weiss Postdoctoral Fellowship. A.A. is supported in part by Bren endowed chair, ONR (MURI grant N00014-23-1-2654), and the AI2050 senior fellow program at Schmidt Sciences.

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

## A    COMPATIBILITY WITH DIFFERENT SELF-SUPERVISED REPRESENTATION LEARNING METHODS

Our framework, as formalized in Eq. (6), enhances model robustness by maximizing the cosine similarity between temporally adjacent points along a deterministic probability flow trajectory. While we implement this objective using the infoNCE loss—a standard choice in contrastive learning (Chen et al., 2021)—our approach is broadly compatible with other self-supervised paradigms, including Joint Embedding Predictive Architectures (JEPA).

Unlike contrastive methods that rely on explicit comparisons between positive and negative pairs, JEPA learns representations by predicting missing information in an abstract latent space, eliminating the need for handcrafted data augmentation heuristics. To integrate JEPA into our framework, we adapt two key components: (1) replacing cosine similarity with Euclidean distance to align with JEPA's emphasis on prediction consistency in representation space, and (2) substituting the contrastive loss with JEPA's predictive loss while reformulating the consistency objective as an MSE loss between positive pairs (constructed as in Section 3). Critically, our framework retains JEPA's core architecture and training designs, requiring only a consistency regularization term. This adaptation preserves JEPA's ability to learn invariant features through latent prediction while inheriting our method's trajectory-aware robustness, demonstrating the flexibility of our approach in unifying contrastive and predictive self-supervised paradigms.

## B    MODEL ARCHITECTURE

We display details of our models in Table 3.

Table 3: Details of rRCM-S, rRCM-B and rRCM-B-Deep.

| Model | #Param | Depth | Dim | MLP Hidden Dim | Output Dim | #Heads |
|---|---|---|---|---|---|---|
| rRCM-S | 25M | 6 | 512 | 2048 | 256 | 8 |
| rRCM-B | 90M | 12 | 768 | 2048 | 256 | 12 |
| rRCM-B-Deep | 177M | 24 | 768 | 4096 | 256 | 24 |

## C    EXPERIMENTAL DETAILS

Table 4: Hyper-parameters used during pre-training.

| Model | Lr | #Iter | Bs | EMA$_1$ | EMA$_2$ | $\tau$ | Optim | Time steps |
|---|---|---|---|---|---|---|---|---|
| *ImageNet* | | | | | | | | |
| rRCM-B | 1e-4 | 600k | 4096 | 0.99 | 0.0 | 0.2 | AdamW | 20 to 80 |
| rRCM-B-Deep | 1e-4 | 600k | 4096 | 0.99 | 0.0 | 0.2 | AdamW | 20 to 80 |
| *CIFAR10* | | | | | | | | |
| rRCM-B | 1e-4 | 300k | 2048 | 0.99 | 0.0 | 0.2 | AdamW | 20 to 80 |

Table 5: Data augmentations utilized when pre-training on ImageNet and CIFAR10.

| Augmentation | Probability $p$ |
|---|---|
| RandomResizedCrop, scale=(0.08, 1.) | 1.0 |
| ColorJitter(0.4, 0.4, 0.2, 0.1) | 0.8 |
| RandomGrayscale | 0.2 |
| GaussianBlur([0.1, 2.0]) | 0.1 |
| Solarize | 0.2 |
| RandomHorizontalFlip | 0.5 |

**Pre-training** During pre-training, we adopt the definition of diffusion models proposed in EDM (Karras et al., 2022) and refer to the implementation of consistency models (Song et al., 2023), including noise schedule, input scaling, time embedding strategy, and time discretization strategy. As for data augmentation strategies, we adopt those utilized in MoCo-v3 (Chen et al., 2021). The temperature value $\tau$ in (9) is set to 0.2 for all experiments. By default, we pre-train rRCM-B and rRCM-B-Deep for 600k steps with a batch size of 4096 on the ImageNet dataset. We pre-train rRCM-B for 300k steps on the CIFAR10 dataset, with a batch size of 2048. Subsequently, we fine-tune our rRCM models separately at various noise levels $\sigma \in \{0.25, 0.5, 1.0\}$. In specific, for both ImageNet and CIFAR-10, we set $\eta_1$ in (12) to 10 at the noise level of 0.25 , and to 20 for noise levels 0.5 and 1.0. In all experiments, $\eta_2$ in (12) is fixed as 0.5.

To enhance training stability, we apply a dynamic EMA schedule for the target model utilized when computing the contrastive loss. Specifically, we gradually increase the EMA rate from 0.99 to 0.9999 following a pre-defined sigmoid schedule, as shown in Figure 6. This schedule is defined by the following equations:

$$l = \sqrt{\frac{k}{K}(E^2 - S^2) + S^2} \tag{13}$$

$$a = \frac{2}{1 + l^{-m\frac{e-S}{E-S}}} - 1 \tag{14}$$

$$EMA = a \cdot E + (1 - a) \cdot S \tag{15}$$

Here, $k$ denotes current training iteration, $K$ is total number of training iteration, $S$ and $E$ represent the start and end EMA rate, $m$ is an empirical parameter, which is set to 10 in our experiments. We present hyper-parameters used in our pre-training experiments in Table 4 and the data augmentation strategies in Table 5.

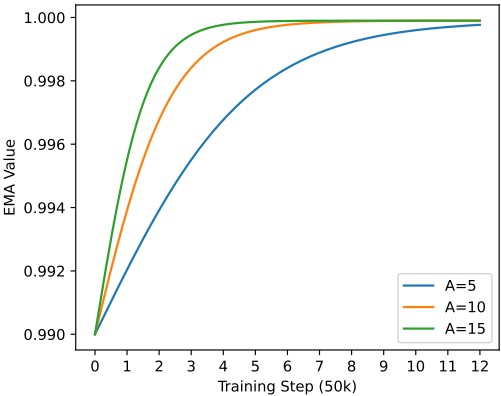

Figure 6: Illustration of the dynamic EMA schedule when changing parameter $m$ from 5 to 15. A larger $m$ corresponds to faster increasing EMA rate.

**Fine-tuning** We fine-tune the pre-trained model following the implementation in (Jeong & Shin, 2020) at three different noise levels $\sigma \in [0.25, 0.5, 1.0]$, and report the best results at each perturbation radius. We tune the pre-trained model for 150 epochs on ImageNet and 100 epochs on CIFAR10.

**Certification** We measure the inference time of all methods on a single A800 GPU. For classical methods, we evaluate with a batch size of 4000 on ImageNet and batch size equals 1000 on CIFAR10. For diffusion-based methods and our rRCM models, we evaluate with a batch size of 100 on ImageNet and 500 on CIFAR10.

**Scalability** After pre-training, we merely fine-tuning the model at noise level $\sigma = 1.0$, and we report the certified accuracy at different perturbation radii.

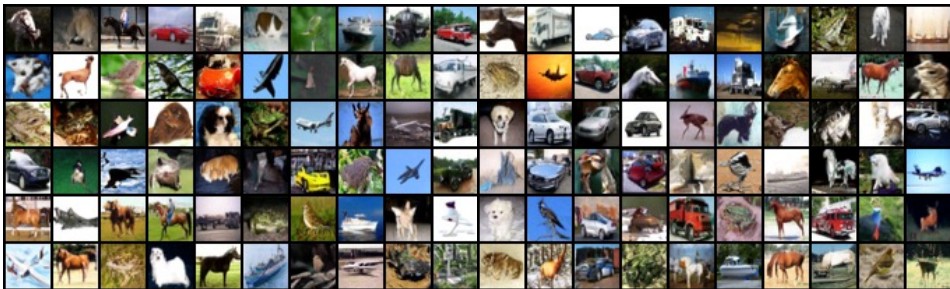

Figure 7: Images generated by conditioning on the output of our rRCM-B model.

## D   QUANTITATIVE ANALYSIS ON THE DEGREE OF REPRESENTATION ALIGNMENT

To further demonstrate the model's ability to align representations by generating meaningful outputs from pure noise inputs, we reuse the rRCM-B model from our CIFAR10 experiment to conduct image generation experiments. In detail, we train a diffusion model conditioned on the output of our rRCM-B model, which takes clean images as input. When generating images, we first generate representations using rRCM-B by feeding in pure noises sampled from the Gaussian prior, defined in Section 2. We then use these representations as conditions to the diffusion model to generate images. As a result, we achieve an FID (Heusel et al., 2017) score of 5.31 measured with 50k generated images. Uncurated image generation results are displayed in Figure 7. We train the diffusion model based on U-ViT-Small (Bao et al., 2023) for 500k steps at a batch size of 128. During sampling, we use DPM-Solver (Lu et al., 2022) to generate images with 50 reverse sampling steps. As conditioning input to the diffusion model, we use features from the MLP head of our rRCM-B model, normalized by their mean and standard deviation.

## E   QUANTITATIVE ANALYSIS OF THE SEMANTIC SIMILARITY BETWEEN POINTS ON DIFFERENT PF ODE TRAJECTORIES

During pre-training, each positive pair is generated from the same clean image perturbed by identical Gaussian noise but at different noise levels. Points in the sample space are treated as solutions of the PF ODE and aligned with their corresponding unique initial point. However, achieving strong certified robustness requires consistent class predictions among points on the stochastic forward trajectory. Specifically, the certification process involves predicting class labels for perturbed samples constructed via the forward SDE, where points are not necessarily confined on the same PF ODE trajectory. Consequently, theses points share similar, rather than identical, semantics to the initial point. As the noise level increases, the semantic similarity between the perturbed and clean images on the stochastic forward trajectory diminishes, which ultimately sets an upper bound on the robustness of our model. This phenomenon, representing a fundamental limitation of all diffusion-based methods, has also been studied in Zhang et al. (2023).

To assess the semantic similarity between points on different PF ODE trajectories, we first train a linear head on clean images using frozen features from the pre-trained rRCM-B model. We then evaluate classification accuracy on noisy samples created by adding varying levels of noise to clean images following the forward SDE of diffusion models. Additionally, we reuse the model in Section D and visualizes images generated by conditioning on representations extracted from points along the stochastic forward trajectory. As shown in Table 6 and Figure 8, increasing noise level leads to a monotonic drop in classification accuracy, with the image content gradually diverging from the original clean image. Furthermore, we report the Fréchet Distance (FD) (Li et al., 2023; Heusel et al., 2017) between representations extracted from $x_{t_N}$ and $x_{t_0}$ in Table 6. A lower RFD value, akin to a reduced FID score (Heusel et al., 2017), indicates greater similarity. This suggests that, despite the differences from their corresponding initial points, the model's predictions on noisy samples still capture meaningful semantics.

Table 6: Quantitative results of the semantic similarity between points on different PF ODE trajectories. Utilizing features from the pre-trained rRCM model, we train a linear head on clean samples while evaluating it on noisy sample of various noise levels.

| Dataset | RFD | Linear Probing Acc % at $\sigma$ | | | |
| --- | --- | --- | --- | --- | --- |
| | | 0 | 0.25 | 0.5 | 1.0 |
| ImageNet | 9.79 | 72.71 | 65.47 | 55.62 | 44.86 |
| CIFAR10 | 3.73 | 87.92 | 72.09 | 65.87 | 50.84 |

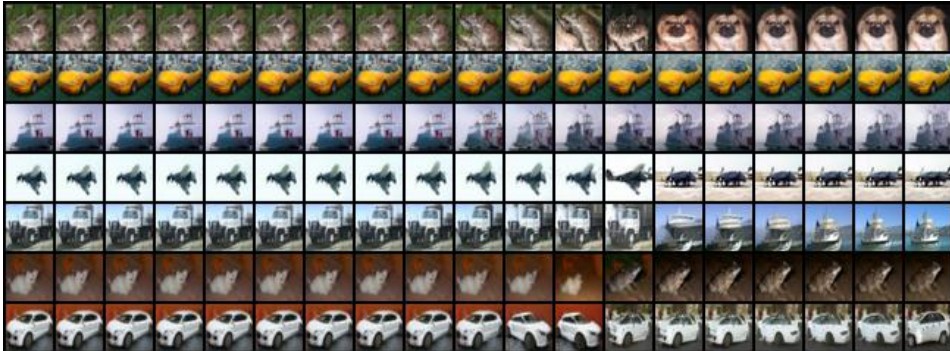

Figure 8: Images generated by conditioning on representations from rRCM-B. The representations are extracted from noisy samples, constructed following forward SDE of diffusion models. From left to right, the time step progressively increases, indicating an increase in the magnitude of noise.

## F COMPARISON WITH NOISE AUGMENTED CONTRASTIVE LEARNING

In this section, we compare rRCM-B with MoCo-v3 on CIFAR10 dataset with performance measured by certified accuracy at various perturbation radii. Specifically, we pre-train a ViT-B model, which has the same amount of model parameters as our rRCM-B model, with MoCo-v3 (Chen et al., 2021) that is additionally equipped with Gaussian noise augmentation. We follow the settings of MoCo-v3 and pre-train the ViT-B model for 300k iterations with a batch size of 256, same as our rRCM-B model. Subsequently, we fine-tune the ViT-B model at noise level $\sigma = 1.0$. In early experiments, we ablate the fine-tuning settings of the ViT-B model and observe similar certified robustness across various configurations. Therefore, we adopt the same fine-tuning settings as our rRCM-B model, including learning rate, data augmentation strategies, training batch size, and total training epochs. As illustrated in Figure 9, the certified accuracy of the ViT-B model is significantly lower than that of our rRCM-B model at all perturbation radii. This highlights that our method, which leverages a structured noise schedule and consistency loss, is fundamentally different from MoCo-v3 which is additionally equipped with Gaussian noise augmentations.

## G ABLATION STUDIES ON HYPER-PARAMETERS

In this section, we ablate our key designs on CIFAR10 dataset. We compare the performance of various settings and report the classification accuracy under different perturbation radii using $N = 100k$ smoothing noises. By default, we pre-train rRCM-B model for 300k iterations with a batch size of 256. For efficiency, we merely fine-tune the pre-trained model at the noise level of $\sigma = 1.0$.

**Ablation on $EMA$ rate and temperature value $\tau$.** We ablate the $EMA$ rate value utilized when computing consistency loss, and ablate the temperature value $\tau$ for both consistency and constrastive loss. We illustrate the results in Figure 10.

**Training on Restricted Noise Levels.** Following (Carlini et al., 2022), we compare five different models pre-trained under restricted noise levels in two distinct settings. (1) Aligning sample points on a partial reverse sampling trajectory: In this experiment, we set $T = 1$ ($T = 80$ in our default

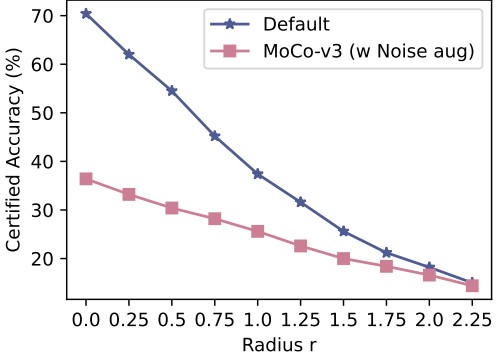

Figure 9: Our method is remarkably different from MoCo-v3 that is additionally equipped with Gaussian noise augmentation. The experiment is conducted on CIFAR10.

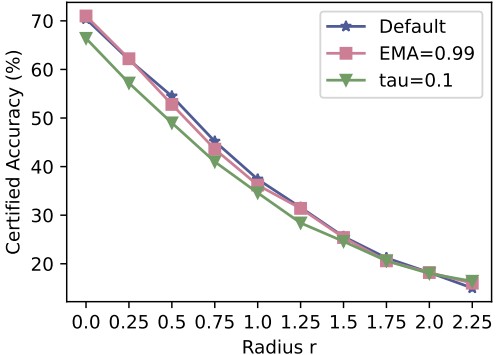

Figure 10: Ablation study on the hyper-parameter settings. By default, we use EMA2=0.0 and $\tau = 0.2$ for computing consistency loss.

setting) as the endpoint, resulting in $t_N = 1$, where $n \in 1, \ldots, N$. (2) Aligning sample points directly with the initial point: Specifically, we select points at three different noise levels along the trajectory: $t_n = 0.5$, $t_n = 1.0$, and $t_n = 2.0$. We present results for the model trained by aligning points at these noise levels with the initial point.

The results are displayed in Figure 11. It is observed that the model, trained by directly aligning points with initial point, yields worse performance as the semantic gap between the two points getting larger.

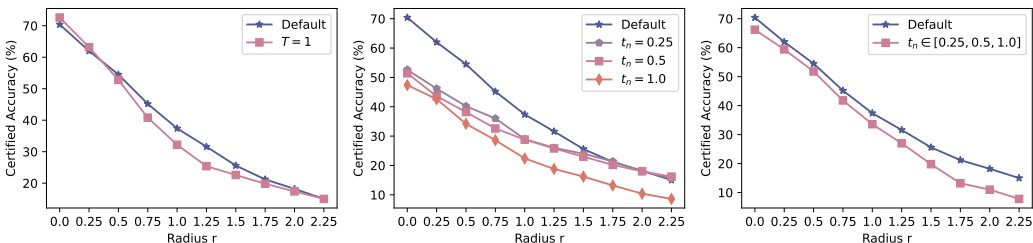

Figure 11: Training on restricted noise levels, including setting the rightmost endpoint at $T = 1.0$ and aligning points with the initial point: $t_n \in 0.5, 1.0, 2.0$ or $t_n \in [0.5, 1.0, 2.0]$.

Table 7: Certified accuracy of rRCM-B on ImageNet under different perturbation radii.

| $\sigma$ | eval at $\sigma$ | Certified Accuracy at $r$ | | | | | |
|---|---|---|---|---|---|---|---|
| | | 0.0 | 0.5 | 1.0 | 1.5 | 2.0 | 2.5 |
| | 0.25 | **0.768** | **0.63** | 0.0 | 0.0 | 0.0 | 0.0 |
| 0.25 | 0.5 | 0.616 | 0.476 | 0.382 | 0.23 | 0.0 | 0.0 |
| | 1.0 | 0.376 | 0.294 | 0.222 | 0.168 | 0.12 | 0.078 |
| | 0.25 | 0.694 | 0.586 | 0.0 | 0.0 | 0.0 | 0.0 |
| 0.5 | 0.5 | 0.672 | 0.566 | **0.456** | 0.346 | 0.0 | 0.0 |
| | 1.0 | 0.494 | 0.392 | 0.322 | 0.266 | 0.218 | 0.156 |
| | 0.25 | 0.674 | 0.554 | 0.0 | 0.0 | 0.0 | 0.0 |
| 1.0 | 0.5 | 0.632 | 0.54 | 0.442 | 0.346 | 0.0 | 0.0 |
| | 1.0 | 0.532 | 0.462 | 0.396 | **0.348** | **0.28** | **0.226** |

Table 8: Certified accuracy of rRCM-B-Deep on ImageNet under different perturbation radii.

| $\sigma$ | eval at $\sigma$ | Certified Accuracy at $r$ | | | | | |
|---|---|---|---|---|---|---|---|
| | | 0.0 | 0.5 | 1.0 | 1.5 | 2.0 | 2.5 |
| | 0.25 | **0.774** | **0.64** | 0.0 | 0.0 | 0.0 | 0.0 |
| 0.25 | 0.5 | 0.692 | 0.552 | 0.43 | 0.32 | 0.0 | 0.0 |
| | 1.0 | 0.502 | 0.412 | 0.338 | 0.258 | 0.202 | 0.138 |
| | 0.25 | 0.718 | 0.612 | 0.0 | 0.0 | 0.0 | 0.0 |
| 0.5 | 0.5 | 0.682 | 0.592 | **0.512** | 0.4 | 0.0 | 0.0 |
| | 1.0 | 0.562 | 0.498 | 0.412 | 0.34 | 0.266 | 0.214 |
| | 0.25 | 0.678 | 0.604 | 0.0 | 0.0 | 0.0 | 0.0 |
| 1.0 | 0.5 | 0.668 | 0.594 | 0.486 | 0.4 | 0.0 | 0.0 |
| | 1.0 | 0.572 | 0.51 | 0.434 | **0.372** | **0.326** | **0.25** |

## H BASELINE METHODS

We compare our method with nine different baseline methods, including: (1) Gaussian (Cohen et al., 2019) trains model with Gaussian noise augmented samples; (2) Consistency (Jeong & Shin, 2020) trains model by additionally regularizing the model output on two Gaussian noise augmented views of the same clean sample; (3) SmoothAdv (Salman et al., 2019a) trains model on adversarial samples crafted during training; (4) Boosting (Horváth et al., 2021) ensembles up to 10 different smoothed classifiers; (5) MACER (Zhai et al., 2020) trains models by directly optimizing for larger certified radius; (6) SmoothMix (Jeong et al., 2021) trains model by on samples created by mixing up adversarial samples and Gaussian perturbed samples; (7) DDS (Carlini et al., 2022) uses a diffusion model to purify perturbed samples, followed by classification with an off-the-shelf classifier; (8) DensePure (Xiao et al., 2022) also incorporates diffusion model with multi-step purification and applies majority voting on class predictions; (9) DiffSmooth (Zhang et al., 2023) uses a diffusion model to purify perturbed samples and employs a smoothed classifier on noisy samples created by adding local smoothing noise to the purified samples, with majority voting for class prediction. We re-implement DensePure by setting the reverse sampling step to 5 as suggested in their work. For DensePure and DiffSmooth, we apply various majority voting numbers, as detailed in Table 1.

## I FURTHER EXPERIMENTAL RESULTS

We show detailed certified accuracy of our models in Table 9, Table 7 and Table 8.

Table 9: Certified accuracy on CIFAR-10 under different perturbation radii.

| $\sigma$ | eval at $\sigma$ | Certified Accuracy at $r$ | | | | |
|---|---|---|---|---|---|---|
| | | 0.0 | 0.25 | 0.5 | 0.75 | 1.0 |
| | 0.25 | **0.836** | **0.734** | **0.614** | 0.458 | 0.0 |
| 0.25 | 0.5 | 0.728 | 0.638 | 0.54 | 0.438 | 0.336 |
| | 1.0 | 0.518 | 0.46 | 0.378 | 0.312 | 0.246 |
| | 0.25 | 0.798 | 0.712 | **0.614** | **0.48** | 0.0 |
| 0.5 | 0.5 | 0.686 | 0.622 | 0.52 | 0.444 | 0.364 |
| | 1.0 | 0.492 | 0.436 | 0.378 | 0.34 | 0.278 |
| | 0.25 | 0.722 | 0.652 | 0.576 | 0.476 | 0.0 |
| 1.0 | 0.5 | 0.618 | 0.556 | 0.5 | 0.444 | **0.392** |
| | 1.0 | 0.5 | 0.446 | 0.408 | 0.356 | 0.296 |

