# OpenReview forum: "Robust Representation Consistency Model via Contrastive Denoising"
_ICLR.cc/2025/Conference — ICLR 2025 Poster_

### Official Review · Reviewer_Svp1 · 2024-10-30

**Soundness:** 3
**Presentation:** 4
**Contribution:** 3
**Rating:** 8
**Confidence:** 3

**Summary:**

The authors propose a novel defense method for image classifiers utilizing contrastive denoising. Experimental results indicate that their approach significantly enhances the model's robustness while reducing inference latency.

**Strengths:**

The authors present strong empirical results, claiming that their method outperforms existing methods in the literature by as much as 5.8% while reducing inference costs by 46 times on ImageNet. The concept of training a robust model using the contrastive denoising method has the potential to benefit adversarial defense society and inspire future research.

**Weaknesses:**

Honestly, I’m not familiar with the denoising-based adversarial defense literature, so my concerns mainly focus on the experimental settings and results.

1. In Figure 1. The authors compare their method with others from the literature. While their method achieves competitive accuracy, the model structure (number of parameters) differs from those used in other studies. It is unclear whether the reduction in inference latency is due to a smaller model size, a high-performance deep learning toolkit (Footnote 5), or advancements in the proposed method. What is the adversarial accuracy of the methods from the literature when using the same models as those employed by the authors in this paper?

2. Line 228-229. Could author further clearify what is role of the teacher model, in the teacher-student training paradigm they used in the paper?

3. What is $Z_1$ and $Z_2$ in the pseudocode of algorithm 1?

4. Line 255-264. Why can a trainable MLP be considered a data augmentation method? Setting data augmentation aside, I still cannot agree with the claim that adding a trainable MLP can "ensure" the model captures meaningful representations. It would be helpful if the authors could provide more evidence to support this statement.

5. In Figure 3, the authors compare their results with those of MoCo V3. However, the accuracy of MoCo V3 is significantly lower than what is reported in the original paper when the radius is set to zero, raising concerns about the correctness of the implementation and the fairness of the comparison.

6. Line 466. Minor: It seems it should be Figure 3 rather than Figure 4.

7. NOT A WEAKNESS (CONCERN). I’m just curious—since the paper is focusing on the defense method, could the authors provide further explanation of the motivation behind Section 4.5, "Image Generation"?

**Questions:**

Please see the weaknesses section.

**Details Of Ethics Concerns:**

No Ethics concerns.

---

> ### Author Response · Authors · 2024-11-23
>
> Thank you for your valuable feedbacks on our experiment results. As suggested, we have polished our writings (weakness 2-4, 6).
>
> ## Part 1/2
> **response to weakness**
>
> - Source of the reduction in inference time
>
> The key idea behind DDS [3], DensePure [4], and DiffSmooth [5] is to purify perturbed samples using an extra diffusion model and subsequently classify the purified results using a separate classification model. When reproducing these methods and reporting their results in Table 1, we use a ViT classifier of the same amount of parameters as our rRCM model. However, during prediction, these methods require an additional diffusion model for purification and rely on majority voting for class prediction, which significantly increases both model parameters and the number of model forward passes. In contrast, our method utilizes a single model and requires only a single forward pass when certified on each sample, attributing the improvement in efficiency primarily to the advancements of our approach.
>
> During the certification process, classical methods also predict class labels of noisy images utilizing a single model within a single forward pass. When comparing with them, we acknowledge that our lower inference latency can be partially attributed to advancements in high-performance deep learning toolkits (as noted in Footnote 5). To provide a fair comparison, we evaluate our method against better-performing diffusion-based methods using similar modern toolkits, achieving significant speedups.
>
> However, as we cannot reimplement all those classical methods and most of them utilize a CNN-like architecture, we cannot precisely quantify how much faster they might become with similar toolkits. While we believe the difference would not be substantial, we included the footnote for transparency. Importantly, even with comparable inference latency, our model demonstrates significantly superior performance, which we attribute to our advanced method and design.
>
> - Certified accuracy of existing methods when utilizing the same model architecture as our rRCM
>
> Considering that there are no available results in the literature for a fair comparison, we re-implement and compare with Gaussian [2] based on a ViT-B architecture, which has the same number of parameters as our rRCM-B model. Please refer to Rebuttal-Table 2 in our general response for more details. Similar to the results presented in Table 1, our model achieves superior results compared to this method.
>
> - Role of teacher model
>
> There are several hypotheses regarding the underlying rationale of the teacher-student training paradigm and the role of the teacher model. For instance, as discussed in Section 5 of SimSiam [9], the teacher model serves to predict the expectation over augmentations. This makes the overall training paradigm analogous to the Expectation-Maximization algorithm.
>
> - Role of MLP
>
> In our previous version, we would like to emphasize that incorporating a trainable MLP head, along with additional data augmentation on the model's input, helps the model capture more effective representations. Empirically, as demonstrated in SimCLR [10] (Figure 8, Appendix B.4), training a contrastive model with a non-linear MLP head results in stronger representations, yielding a 3% improvement in classification performance compared to a linear head. The use of a non-linear MLP head has become a standard practice in the contrastive learning literature [10-12]. Consequently, we also adopt the MLP head in our method. We updated relevant descriptions (see also Section 3.3) to avoid any misunderstanding.

---

> ### Author Response · Authors · 2024-11-23
>
> ## Part 2/2
> - Accuracy of MoCo-v3 model is abnormally low
>
> To demonstrate that our method is fundamentally different from conventional contrastive learning approaches that incorporate Gaussian noise augmentation, we conducted a comparison with MoCo-v3 [11], training it with additional random Gaussian noise augmentation. Specifically, we trained MoCo-v3 on noisy images with a noise level of 1.0. As expected, since MoCo-v3 is not designed for learning robust representations, this training strategy degraded the quality of the learned representations. Nevertheless, despite the performance drop, it improved the model’s certified robustness compared to training on clean images alone.
> Additionally, following the evaluation protocol in Certified Robustness (CR), we certified MoCo-v3’s accuracy on noisy samples (noise level = 1.0) during evaluation. This implies that the certified accuracy at radius = 0 is not directly comparable to the classification accuracy on clean images. Consequently, the (certified) classification accuracy we report is noticeably lower than the numbers reported in the original MoCo-v3 paper.
> In early experiments, to ensure the correctness of our implementation, we have conducted a quick reproduction experiment with suboptimal hyperparameters and achieved 74.5% linear probing accuracy—2.2% lower than the results reported in the original MoCo-v3 paper. This outcome provides additional confidence in the validity of our implementation.
>
> - Motivation behind image generation
>
> We refer to our general response for a detailed explanation. In short, we wanted to show that our trained model can *generate* meaningful representations (see also Tab. 6, where we measure the Fréchet distance between generated representations and clean representations, both extracted with our rRCM-B model) that can, e.g., be used for image generation downstream tasks. Moreover, these representation generation capabilities validate that our training successfully maps points on the ODE trajectories to the same point in the latent space.

---

> > ### Comment · Reviewer_Svp1 · 2024-11-27
> >
> > "The key idea behind DDS [3], DensePure [4], and DiffSmooth [5] is to purify perturbed samples using an extra diffusion model and subsequently classify the purified results using a separate classification model. When reproducing these methods and reporting their results in Table 1, we use a ViT classifier of the same amount of parameters as our rRCM model. However, during prediction, these methods require an additional diffusion model for purification and rely on majority voting for class prediction, which significantly increases both model parameters and the number of model forward passes. In contrast, our method utilizes a single model and requires only a single forward pass when certified on each sample, attributing the improvement in efficiency primarily to the advancements of our approach." Please revise the caption of the figure, to make it clear.
> >
> > "To demonstrate that our method is fundamentally different from conventional contrastive learning approaches that incorporate Gaussian noise augmentation, we conducted a comparison with MoCo-v3 [11], training it with additional random Gaussian noise augmentation. Specifically, we trained MoCo-v3 on noisy images with a noise level of 1.0. As expected, since MoCo-v3 is not designed for learning robust representations, this training strategy degraded the quality of the learned representations. Nevertheless, despite the performance drop, it improved the model’s certified robustness compared to training on clean images alone. Additionally, following the evaluation protocol in Certified Robustness (CR), we certified MoCo-v3’s accuracy on noisy samples (noise level = 1.0) during evaluation. This implies that the certified accuracy at radius = 0 is not directly comparable to the classification accuracy on clean images. Consequently, the (certified) classification accuracy we report is noticeably lower than the numbers reported in the original MoCo-v3 paper. In early experiments, to ensure the correctness of our implementation, we have conducted a quick reproduction experiment with suboptimal hyperparameters and achieved 74.5% linear probing accuracy—2.2% lower than the results reported in the original MoCo-v3 paper. This outcome provides additional confidence in the validity of our implementation."It is unclear whether the parameter is biased toward the authors' settings. Please clarify this in the paper.
> >
> > "We refer to our general response for a detailed explanation. In short, we wanted to show that our trained model can generate meaningful representations (see also Tab. 6, where we measure the Fréchet distance between generated representations and clean representations, both extracted with our rRCM-B model) that can, e.g., be used for image generation downstream tasks. Moreover, these representation generation capabilities validate that our training successfully maps points on the ODE trajectories to the same point in the latent space."
> > It still didn’t make much sense. Again, I want to clarify that this is not a criticism. Even if the processed image is highly distorted and unrecognizable to humans, as long as the DNN classifier can correctly classify it, it is acceptable.
> >
> > I might increse the rating if aforementioned questions has been solved

---

> ### Author Response · Authors · 2024-11-29
>
> Thank you for your valuable feedback on our rebuttal, which helped us to further improve our work! We have revised our paper based on your suggestions, clarifying the model settings used during certification in the caption of Figure 2 and providing detailed explanations of the experimental settings used to compare with MoCo-v3 (Appendix, lines 857–863). Please let us know if we should make further edits in our final version.
>
> **Experiment settings when comparing with MoCo-v3**
>
> In the experiment comparing with MoCo-v3, we first pre-train a ViT-B model (with the same amount of model parameters as rRCM-B) following MoCo-v3's settings. For finetuning, we use the tuned hyperparameters (e.g., learning rate, batch size, and data augmentation strategies) for the ViT-B model, noting that we observed negligible differences in certified accuracy across various fine-tuning configurations. This clarification has been included in our Appendix (lines 857–863).
>
> **Why image generation experiment**
>
> We agree with your perspective that the image generation experiment is not strongly correlated with the primary focus of this work, i.e., certified robustness. We have thus moved this experiment to the Appendix.
>
> The goal of the experiment is to validate the effectiveness of our consistency objective during pretraining, ensuring that the distribution of $f(x_T)$, $x_T \sim \mathcal{N}(0, T^2 I)$, aligns with the distribution of $f(x_0)$, $x_0\sim p(x_0)$. In other words, the distribution of our model evaluated at pure Gaussian noise approximates the distribution of our model evaluated at the data distribution. In that sense, we demonstrate that our pre-training enables the acquisition of generative capabilities (in latent space) using solely a discriminative optimization objective (contrastive learning). Beyond the scope of certified robustness, our method thus serves as a preliminary exploration of learning both strong representations and their corresponding distribution in the latent space in a way that is analogous to denoising training in the pixel space.
>
> ---
>
> Again, we thank you for your constructive and valuable suggestions to our work. Please let us know if you have any further questions or concerns. If our responses have adequately addressed your concerns, we kindly ask you to consider updating your score.

---

> > ### Comment · Reviewer_Svp1 · 2024-12-01
> >
> > All my concerns have been addressed, and the mark has been increased. Thank you to the authors for clarification.

---

### Official Review · Reviewer_wTyH · 2024-10-30

**Soundness:** 3
**Presentation:** 3
**Contribution:** 3
**Rating:** 8
**Confidence:** 4

**Summary:**

This paper proposes to incorporate a new diffusion trajectory based consistency loss to the self-supervised pretraining of vision transformers for robust representation learning. The resulting model can be utilized directly in the randomized smoothing framework for certified adversarial robustness, removing the need of pretrained diffusion models employed by prior works. The proposed method, rRCM, is competitive with prior art at low perturbation radii with lower latency, while outright exceeding previous state of art performance at larger perturbation radii.

+++++++++++++++++++++++++++++++++++++++++++++++++++++

Post rebuttal edit:

Most of my concerns regarding clarity and presentation have been addressed in the revision. I think the paper is in solid shape and hence would recommend accepting for publication.

**Strengths:**

1. The proposed method makes sense intuitively and seems like a novel way to bake adversarial robustness into representation learning pipelines
2. Experimental evaluations are fairly thorough and the performance improvements over prior methods are quite large.

**Weaknesses:**

1. I can’t quite understand what is the message behind the image generation experiment in section 4.5. Is the diffusion model trained on actual images in CIFAR10? If so, then regardless of whether the conditioning encoder extracts meaningful semantic representations, the diffusion model will produce images that resemble CIFAR10 images.

2. Overall, writing could use some polish. Some examples:
  * Line 42 “While empirical defenses train DNNs to be robust to known adversarial examples (Madry et al., 2017) during training”
  * Line 305 “... . And we…”
 * Line 315 “During pre-training, we adopt the settings of the diffusion model in EDM”. Which settings are you adopting?
* Table 1. Table caption should appear on top of table. Ties should be highlighted (SmoothAdv at r=2.5)
*There is a lot of empty space around figures and some sections in the text that could be removed (such as that on image generation) to make space for a method figure in the main text (but leave out the patch illustration as I think most audiences are familiar with how ViTs work).
* Figures 1,3,4,5 have “classification accuracy” on y-axis, should this be “certified accuracy”?
* Figure 3 is never mentioned in text. Captions or legends in figures 3,4, and 5 should indicate which dataset the evaluation is from.
3. Writing in the method section in particular lack clarity in several places:
* a. While I understand the idea the authors are attempting to convey with equation 6 and 7, their presentation diminishes the rigor of the paper as the equality constraint between $f(x_*)$ in 6 and the dot product being equal to 1 are strong statements. It would be more appropriate to instead state, for instance, that you intend to minimize the differences between two $f(x_*)$ , i.e. $\min_\theta ( f_\theta (x_*) - f_\theta (x_**) )$. As a side note, one clearly does not intend to have equation 6 hold all the way to $t_N$, as the marginal of the PF-ODE at $t_N$ attains a standard Gaussian distribution, such that all trajectories are indistinguishable in distribution at this time step, thereby forcing f to loose all discriminative power.
* b. How is teacher-student training actually related to your proposed method? Is it just the use of the ema weights in the contrastive loss? This seems like a weird point of emphasis as even the SimCLR paper doesn’t consider itself an example of knowledge distillation. The paragraph on teacher-student learning in related works feels similarly ill-named.
* c. In Algorithm 1, it is not clear whether mlp(g( , )) corresponds to the linear head or the non-linear projector head. It is also not clear what ct_loss and cnsis_loss meant (also, there’s a typo “cnsis_loss” becomes “cnsis” on the next line)
* d. Line 247-line 253 cites SimCLR but there doesn’t seem to be any statement in SimCLR that supports the statement of line 247-line 253
* e. Line 255-line 272 was very confusing until I read line 273 stating that you use contrastive loss in addition to the PF-based consistency loss. It would have been much more clear to state that these two losses were used before going into details such as which samples are augmented and the bottleneck layer. Furthermore, an expression of the consistency loss should be included in the methods section for completeness.
* f. There is no subsection on finetuning?
* g. Why the obsession with comparisons to MoCo-V3 but not have it in the main results tables?

**Questions:**

1. Please address my concerns in the weaknesses section. I think the technical content of the paper is great but the presentation of the paper requires significant work.
2. On what hardware configuration and at what batch size are the latency number obtained?
3. What are the “raw” classification accuracies of rRCM-B-Deep on Imagenet at various perturbation radii? (the numbers in bracket in Carlini2022’s table 1 and 2)

---

> ### Author Response · Authors · 2024-11-23
>
> Thank you for your recognition of our work and your constructive feedback. As suggested, we have polished our writings (weaknesses 2, 3.a, 3.c-3.e) and largely revised our presentation of the methodology section, including re-writing the problem formulation in a more rigorous way (see Section 3.2), updating our pseudocode in Algorithm 1, presenting our pre-training method in a clearer way, and present more details and discussions of our fine-tuning method. We have also added two figures (Figures 2 and 3) to provide an overview of our pre-training strategy and the model forward pass.
>
> **response to weakness**
>
> - Why image generation experiment
>
> We refer to our general response for a detailed explanation. In short, we wanted to show that our trained model can *generate* meaningful representations (see also Tab. 6, where we measure the Fréchet distance between generated representations and clean representations, both extracted with our rRCM-B model) that can, e.g., be used for image generation downstream tasks. Moreover, these representation generation capabilities validate that our training successfully maps points on the ODE trajectories to the same point in the latent space.
>
> - Why emphasize teacher-student training paradigm in related work
>
> Our method is inspired by both contrastive learning and consistency models but differs from each in a significant way. The shared connection lies in the teacher-student training paradigm, which has proven effective in both generative and discriminative modeling. Since our method unifies these two aspects by jointly learning (1) denoising (a generative task) and (2) meaningful representations (a discriminative task), we framed our discussion of its relationship with contrastive learning and the consistency model from the perspective of the teacher-student training paradigm.
>
> - Why comparing with MoCo-v3
>
> We include a comparison with MoCo-v3 [11] as an ablation study to show that our method differs significantly from conventional contrastive learning approaches. In particular, we demonstrate that the performance improvements are largely due to our consistency objective (and not just due to contrastive learning). Since MoCo-v3 is neither designed for learning robust representations nor widely used in the relevant literature on certified robustness, we chose not to include it in our main experiments. In our updated paper, we defer the experiment to the appendix as it is not the primary focus of our work.
>
> **response to questions**
>
> - Hardware configuration
>
> Please refer to our general response for detailed information.
>
> - raw classification accuracy
>
> The “raw” classification accuracies of rRCM-B-Deep in Table 1 are shown below (numbers in brackets):
>
> Model Name &nbsp;&nbsp;&nbsp;&nbsp;&nbsp;&nbsp;&nbsp;&nbsp;&nbsp;&nbsp;&nbsp;&nbsp;&nbsp;&nbsp;&nbsp;&nbsp;&nbsp;&nbsp;&nbsp;&nbsp;&nbsp;&nbsp;&nbsp;&nbsp;&nbsp;&nbsp;&nbsp;&nbsp;&nbsp;&nbsp;&nbsp; Certified accuracy at r (%)
> | | 0.5 | 1.0 | 1.5 | 2.0 | 2.5 |
> | --- | --- |  --- |  --- |  --- |  --- |
> | rRCM-B-Deep | 64.0 (77.4) | 51.2 (68.2) | 40 (66.8) | 32.6 (57.2) | 25.0 (57.2) |
>
> ---
>
> Please let us know if you have further questions or concerns. If our responses have adequately addressed your concerns, we kindly ask you to consider updating your score.

---

### Official Review · Reviewer_VAKb · 2024-10-31

**Soundness:** 2
**Presentation:** 2
**Contribution:** 3
**Rating:** 6
**Confidence:** 3

**Summary:**

This paper proposes a new method to improve the robustness of deep neural networks against adversarial perturbations. By reformulating the generative task in consistency models as a discriminative task in latent space, the authors use instance discrimination to ensure consistency of noisy samples along PF-ODE trajectories. This approach enables one-step denoising and classification, reducing computational overhead while achieving a balance between robustness and efficiency across different perturbation radii. Experiments show that the method performs well on large-scale datasets like ImageNet and significantly reduces computational costs compared to traditional methods.

**Strengths:**

+ The overall structure and writing approach of the paper are well-organized and clear.
+ Combining consistency model with robust representation learning is interesting and novel to me. The use of PF-ODE to build a consistent relationship between clean data and perturbed data has a theoretical basis.
+ The proposed method is significantly more time-efficient than previous diffusion-based methods, both theoretically and practically..

**Weaknesses:**

- Some details of the method are unclear; it would be helpful to provide the complete loss function formula used in the approach, along with a detailed explanation.
- The improvements in Certified Accuracy shown in Table 1 are marginal, suggesting limited gains in robustness compared to prior methods.

**Questions:**

- What are z_0 and z_1 in Algorithm 1? More explanation and a detailed description of the proposed method are needed.
- It is still unclear what the upper limit of this method is. I hope the authors can explain this and clarify why the method has or has not reached this limit if applicable.
- It appears that this method relies on the consistency model to denoise perturbed data. My question is why the consistency model can recover perturbed data to the corresponding original data. Although this is not the main contribution claimed by the paper, I still hope the authors can provide a reasonable explanation for this.

---

> ### Author Response · Authors · 2024-11-23
>
> We thank the reviewer for the valuable comments regarding our presentation and our method. We have reorganized the methodology section to include the complete loss function in Eq.7 followed by detailed explanations. We have also updated Algorithm 1 and add relevant descriptions, making it more clear and easier to understand.
>
> **Response to weakness**
>
> In essence, our loss function consists of two components: a consistency loss and a contrastive loss. With consistency loss alone, the model would tend to output trivial and similar representations on distinct points in the sample space, particularly on the initial points. As a result, during training, the model would struggle in differentiating points sampled from PF ODE trajectories, leading to training difficulty.
>
> Therefore, the contrastive loss is primarily introduced to enhance the model’s ability to extract meaningful and unique representations from each clean image, further differentiating points and the corresponding PF ODE trajectories in the sample space. Meanwhile, as demonstrated in the experiment that compares rRCM with MoCo-v3 [11] (deferred to Appendix), the contrastive loss alone is also not sufficient for reaching an outstanding result.
>
> **Response to questions**
>
> - Upper limit of our method:
>
> Intuitively, the upper limit of our method is constrained by the capability of the Consistency Model (CM) framework to preserve semantic similarity between denoised perturbed samples and their corresponding clean images. Specifically, if CM fails to maintain this semantic consistency, our method is also likely to misclassify the perturbed samples. However, we also empirically observe that the performance of our method has not yet plateaued. With more training budget, our method can achieve stronger performance. We add a corresponding explanation in Section 3.4 (Fine-tuning).
>
> - It appears that this method relies on the consistency model to denoise perturbed data. why the consistency model can recover perturbed data to the corresponding original data.
>
> During pre-training and fine-tuning, our model is trained in an end-to-end manner. Afterward, during certification, our method does not require any extra denoising model (e.g., consistency model or diffusion model) to purify perturbed samples. In essence, we certify the robustness of our rRCM model by directly predicting the class label of noisy images within a single model forward pass. We have updated and added more explanations of our pre-training and fine-tuning methods in our paper.
>
> Besides, regarding CM itself, it doesn’t guarantee that perturbed samples are mapped back to their original clean image. This is because the perturbed samples are generated by adding noise to clean images, following the forward process of CM. However, CM only ensures the points on the same Probability Flow (PF) ODE sampling trajectory are mapped to the same initial point, not necessarily for points along the stochastic forward trajectory. As a result, CM could “erroneously” map perturbed samples to a clean image different from their origin at certain noise levels. This phenomenon (also studied in DiffSmooth [5]) explains why both CM and our method could misclassify perturbed samples in such cases.

---

### Official Review · Reviewer_iF9T · 2024-11-03

**Soundness:** 2
**Presentation:** 2
**Contribution:** 2
**Rating:** 5
**Confidence:** 3

**Summary:**

The paper presents the Robust Representation Consistency Model (rRCM), an approach to improve robustness in deep neural networks by leveraging contrastive denoising within diffusion models. Instead of standard generative denoising, the authors reframe denoising as a discriminative task in latent space, enabling implicit one-step classification with reduced computational overhead. The authors perform experiments on datasets like ImageNet and CIFAR10 and investigate the method’s certified accuracy and scalability across perturbation radii, claiming a notable reduction in inference costs.

**Strengths:**

- The paper introduces a structured noise schedule for diffusion-based Randomized Smoothing, reformulating de-noising from a generative to a discriminative task. This is an interesting idea, especially in adversarial robustness, where noise schedules and latent consistency are less commonly used.
- The authors perform a thorough evaluation of their approach, comparing it against multiple state-of-the-art methodologies, in terms of certified accuracy and inference time.

**Weaknesses:**

- Motivation: the main idea of Carlini et al. (2022) is to utilize an existing diffusion model for de-noising, thus getting randomized smoothing "for free", without needing to fine-tune the base classifier. This work reformulates the diffusion process to be more aligned in the latent space, but in the end, we get a classifier that's trained from scratch, with a diffusion-like objective. Why do all that and not just train a classifier on noisy images (since we're going to train in any case)? What is the benefit? The motivation of the approach is not very clear to me. In particular, I cannot follow the diffusion formulation and motivation precisely.
- Presentation: related to the above, the methodology section of the paper is not easy to follow, e.g. there are various details where I don't fully understand how it all comes together (normalization, consistency loss, etc.). It would help if the authors (1) state the high-level motivation of their approach, (2) add clean pseudo-codes showing how the overall process works. Also, figure 2 would benefit from further elaborations.
- Complexity: the approach is technically complex (structured noise schedule, alignment with PF ODE trajectories) which could hinder it's extension in other setups.
- Some things, like e.g. the 46x speed improvement are over-claimed, for instance this holds only for a particular case. On average, the approach is faster than e.g. the Gaussian baseline or Carlini et al. (2022), but not by that much.

**Questions:**

My questions are mainly related towards addressing the weaknesses mentioned before:
- Can the authors clearly motivate the methodology, why this particular approach and not something else? Why e.g. not just train a ViT on a lot of noisy samples?
- Can the presentation be improved so that readers can easily see the overall motivation, and follow along with the details?
- Despite the complexity overhead, does the approach generalize on other e.g. models or datasets, and how easily?

---

> ### Author Response · Authors · 2024-11-23
>
> Thank you for your review and the valuable suggestions regarding our motivation and writing. We have carefully revised our paper based on your feedback. In particular, we explain in our revised version why our method is preferable compared to classical methods that directly train models on noisy samples (lines 87-90, 111-122). To demonstrate this, we reimplemented the Gaussian [2] and present detailed results in our general response, Rebuttal-Table 2. Besides, we updated the problem formulation in Section 3.2 and the pre-training method in Section 3.3. We also added two figures (Figures 2 and 3) and more explanations in Section 3.3 to provide an overview of our method. Moreover, we updated the pseudocode in Algorithm 1 along with explanations (lines 284-297), making it clearer and easier to understand.
>
> ## Part 1/2
> **Response to Weakness**
>
> - Motivation.
>
> Classical methods train models directly on noisy samples but primarily rely on heuristic strategies for choosing the noise, limiting their ability to achieve higher levels of certified robustness. As displayed in Table 1, our method outperforms the Gaussian [2] and Consistency [8] methods by a large extent. Meanwhile, for a fair comparison, we reimplement the Gaussian using a ViT-B model that has the same architecture and number of parameters as our rRCM-B model. As demonstrated in Rebuttal-Table 2 in the general response, our approach still achieves superior certified accuracy, showing that our improvements originate from our training methodology and not merely architectural changes. Furthermore, our method also demonstrates strong scalability when tested on ImageNet. These results underscore the importance of leveraging ideas from consistency models to establish connections between noisy and clean samples, making our method more accurate and efficient than previous approaches. We added more explanations in our introduction in line 87-90 and line 111-122.
>
> - Complexity of our method
>
> Our goal of aligning points on the ODE trajectory (outlined in the revised Section 3.2) motivates why we formulate our training objective as InfoNCE loss. However, different than in standard contrastive learning, the structured noise schedule governs how noisy input data pairs are created for computing the positive sample pairs. These are the two primary design principles of our method. Empirically, as outlined in our pseudocode, only a few lines of code modification are required based on the consistency model's official repository [9]. Besides, our method works seamlessly with vanilla ViT models, eliminating the need for any network modifications and making it easy to scale our method. In addition, combining ideas from consistency models and contrastive learning, our approach demands no additional mathematical foundation beyond what is needed to understand diffusion models and representation learning.
>
> - Inference latency of our method
>
> In terms of inference efficiency, we focus on comparing with diffusion-based methods that previously achieved state-of-the-art certified accuracies. The 46-times reduction in inference time is reported by comparing our method (6s) with DiffSmooth [5] (m=15, 5min 35s) when utilizing 10,000 randomized smoothing noises, as shown in Table 1. When comparing with DensePure [4] (K=1) and DDS [3], our method reduces the inference time by approximately 171 and 39 times, respectively. Notably, we did not report the maximum reduction in inference time compared to DensePure due to the latter's extremely costly certification process. We report the detailed configurations on how we measure the inference time in our general response.

---

> ### Author Response · Authors · 2024-11-23
>
> ## Part 2/2
> **Response to Questions**
>
> - Generalizability to other datasets or model
>
> We conducted additional experiments on the KatherColon [1] dataset, with details presented in our general response. As shown in the Rebuttal-Table 1 in the general response, our pre-trained model demonstrates strong transferability to other datasets with minimal effort.
>
> While ResNet and ViT variants dominate the image classification domain, the success of diffusion models is primarily attributed to architectures like U-Net, DiT, and U-ViT. These two families of model architectures differ greatly in their design. For our method to succeed, we require a general architecture that is compatible with both image recognition and generation tasks.
>
> For ViT-like architectures, we observed in our early experiments that the vanilla ViT model meets our requirements without incurring additional costs. Specifically, we treat both the time condition and the learnable class as input tokens to the ViT model and train it using the InfoNCE-like loss (We present an illustration and detailed explanation of the model forward pass in Figure 3 and Section 3.3, line 276-282). Our method integrates seamlessly with the original ViT architecture, eliminating the need for any network modifications.
>
> On the other hand, to the best of our knowledge, no CNN-based architecture is inherently compatible with both image recognition and generation tasks. Therefore, we did not implement our method on CNN-like architectures. Designing a novel CNN architecture suitable for both tasks is a non-trivial challenge that lies outside the scope of this work. We kindly ask the reviewers not to consider this as a weakness of our method.
>
> ---
>
> Please let us know if you have any further questions or concerns. If our responses have adequately addressed your concerns, we kindly ask you to consider updating your score.

---

> ### Author Response · Authors · 2024-11-25
>
> Dear Reviewer iF9T,
> As the discussion period is approaching its end, we would like to kindly ask whether we have adequately addressed your concerns. If there are any remaining issues, we would greatly appreciate it if you could share them with us, so we have enough time to provide a comprehensive response.
> Thank you once again for your time and valuable feedback!

---

> > ### Comment · Reviewer_iF9T · 2024-11-25
> >
> > Thank you very much for your detailed response, and for taking my comments into account! I went through the new version and your rebuttals, and still have the following concerns:
> > - Presentation: despite the changes it's still not easy for me to follow every step in the methodology, and the motivations behind them.
> > - High-level idea: unfortunately, I still cannot precisely understand the motivation of your method. Perhaps it could be because I'm not so familiar with that space. Could you please clarify this further? Can be done also in an Appendix or in comments, so that you don't have to modify the main text again.
> > - Comparison with vanilla ViT: Your ViT in rebuttal table 2 seems under-trained to me: it achieves an accuracy of only 41% at radius 0 (e.g. noise-less case). Why is that? This can raise suspicions about the fairness of the comparison. Also, why do you claim that the training method of e.g. (Cohen et al., 2019) is based on heuristics? I mean, to get a good smooth classifier, we need to make it so that it classifies noisy images the same as before, and this is what they train it to do - why is that a heuristic?
> > - Latency: can you clarify from where the latency gains of your method stems from? For example, your method and (Carlini et al., 2022) both use a similar architecture, why is your method faster?
> > - Code: could you release your code in an anonymized repository or zip file to take a look? Perhaps this could help me further understand some details.
> >
> > Overall, the experimental results look good, it's mostly a presentation issue for me. Please take the time to improve it to a point so that people not so familiar with the topic can follow and appreciate your contribution! Thank you again, and apologies for the many comments!

---

> ### Author Response · Authors · 2024-11-29
>
> # Part 1/4
>
> Thank you for your prompt reply! We sincerely appreciate your valuable feedback on the presentation and motivation of our method! We believe it’s important to present our work clearly in a way that researchers new to this field can easily follow. In this reply, we primarily present more high-level and intuitive explanations of our idea along with the underlying rationales of the key designs of our methodology. To improve the accessibility of our work, we will add this discussion to our appendix in the final version. In addition, we also uploaded our code to help you better understand the details of our method. We will incorporate and refine these aspects in our final version.
>
> **Motivation of our work**
> Randomized smoothing is a technique that provides a theoretical guarantee for the robustness of arbitrary classifiers against adversarial perturbation. Specifically, given a base classifier $f$, it constructs a smoothed classifier $F$, defined as
> $$ F(x) = \operatorname{argmax}\_{c \in \mathcal{Y}} \\; \mathbb{P}\_{\epsilon \sim \mathcal{N}(0, I)}(f(x+\sigma\epsilon) = c ), \quad (Rebuttal-Eq-1)$$
> Intuitively, for each clean image $x$, the smoothed classifier outputs the most probable class by majority voting over the class predictions of the base classifier, using a large number (e.g., 100,000, as is common practice) of perturbed samples $x+\sigma\epsilon$ as input. These perturbed samples are generated by adding noise, independently sampled from a Gaussian distribution $\mathcal{N}(0, \sigma^2I)$, to $x$. Notably, during certification, the final class label of each test sample is determined utilizing the smoothed classifier $F$. The certification process, relying on majority voting over noisy samples, fundamentally differs from the evaluation procedure of a typical image classification task, where the base classifier is directly evaluated.
>
> Afterward, one can compute the certified radius on each test sample using the prediction margin of the smoothed classifier $F$. Specifically, given the probabilities $p_A$ and $p_B$ of the most probable class A and the runner-up class B output by the smoothed classifier $F$ on a test sample $x$, we compute the certified radius $R(x)$ using Equation (5). The certified radius guarantees that $F$ classifies all perturbations of $\ell^2$-norm smaller than $R(x)$ around $x$ as class A (see [2, Thm. 1]). Finally, we report the fraction of correctly predicted samples with a radius above a certain threshold (in our paper, we choose thresholds $0$ to $2.5$ in steps of $0.5$, following the literature). Specifically, for a threshold $r \ge 0$, the certified accuracy is computed as
> $$ACC^r_{\text{certified}} = \frac{1}{N} \sum_{n=1}^{N}{[R(x_n) \ge r]}, \quad (Rebuttal-Eq-2)$$
> where $N$ is the size of the test dataset used for certification and $[R(x_n) \ge r]$ denotes the Iverson bracket, i.e., takes the value $1$ if the certified radius (computed by Equation (5) in the paper) for the $n$-th clean sample $x_n$ is larger or equal to $r$ and $0$ else. We note that the certified accuracy at all thresholds is based on the classification of *noisy* samples at a specific noise level $\sigma$ (e.g., for $r=0$, the noise level is usually at $\sigma=0.25$).
>
> To date, the most popular randomized smoothing methods can be roughly divided into two categories: classical methods and diffusion-based methods. The first ones optimize models for stronger robustness by directly training them on noisy samples, which are constructed either by adding Gaussian noise [2, 8, 13, 14, 15] or by applying adversarial perturbations (crafted during training) [13] to clean images. On the other hand, diffusion-based methods [3, 4, 5] solve the problem in two stages by utilizing an off-the-shelf diffusion model $D$ to purify the noisy input followed by a classifier $g$, trained on clean images, to predict the class label on the purified results. Consequently, the base classifier is given as $f=g\circ D$. However, both these approaches suffer from several limitations.
>
> First, directly training models with standard classification objectives on noisy samples is ineffective for achieving strong robustness. At large noise levels, where the perturbed samples contain limited semantics, the model struggles to learn general and robust representations without appropriate guidance or information. Therefore, classical methods like [2, 15] achieve low certified accuracy on large perturbation radii (see also Table 1). To mitigate this, other classical methods [8, 13, 14, 15] introduce various techniques, including consistency regularization [8] and adversarial training [13, 14]. However, these methods suffer from other drawbacks, including training instability [8] and high training costs [13, 14], making it difficult to scale them effectively and achieve strong robustness.

---

> ### Author Response · Authors · 2024-11-29
>
> # Part 2/4
>
> Meanwhile, diffusion-based methods [3, 4, 5] achieve superior certified accuracy and can further improve model robustness at the cost of inference latency. As aforementioned, within each forward pass of the base classifier $f = g \circ D$, such methods first purify the perturbed samples utilizing a diffusion model $D$ and then classify the purified results with the classification model $g$. To improve performance, multiple denoising steps are applied during purification. Moreover, by utilizing different random seeds, the purification is repeated several times on each noisy sample $x+\sigma \epsilon$, generating multiple purified samples that are subsequently classified by $g$. These classifications undergo an additional majority voting step to yield the final prediction of the base classifier. It is important to note that this majority voting occurs "within" the base classifier and differs from the majority voting employed by the smoothed classifier $F$. Specifically, in this case, $F$ can be expressed as
> $$ F(x) = \operatorname{argmax}\_{c \in \mathcal{Y}} \\; \mathbb{P}\_{\epsilon \sim \mathcal{N}(0, I)}\Big(MV\big( g(D( x+\sigma\epsilon; S_1)), g(D( x+\sigma\epsilon; S_2)), ... \big)=c\Big). \quad (Rebuttal-Eq-3)$$
> Here $MV$ denotes the majority voting operation, $S_k$, $k\in\\{1, ..., K\\}$, is the random seed used in the diffusion model $D$ for purification, and $g$ is the classification model.
>
> While such approaches achieve state-of-the-art robustness guarantees, they result in significantly higher inference costs. For instance, as reported in our paper, the inference time for [4] can reach up to 52 minutes per image, using $K=5$ seeds and utilizing $5$ denoising steps in the diffusion model, resulting in a total of 30 forward passes (25 for the diffusion model $D$ and 5 for the classifier $g$) within the base classifier $f$ (which needs to be evaluated for 100,000 perturbations of an image to obtain the output of the smoothed classifier $F$). Such high computational costs make diffusion-based methods impractical for deployment in real-world security-sensitive applications where low latency is critical.
>
> In conclusion, classical methods are efficient (the employed base classifier $f$ is a classification model that predicts class labels of noisy samples within a single forward pass) but suffer from fundamental training difficulties [8, 13, 14], preventing them from reaching strong robustness at all perturbation radii. On the contrary, diffusion-based methods achieve superior model robustness yet at a formidable cost of high inference latency. Our method bridges the gap between efficiency and performance in the field of certified robustness, achieving superior performance at all perturbation radii while maintaining inference costs similar to classical methods. In particular, our method outperforms state-of-the-art diffusion-based methods while requiring a single forward pass for each prediction made by $f$.

---

> ### Author Response · Authors · 2024-11-29
>
> # Part 3/4
> **Rationale of the key designs of our method**
>
> *A high-level explanation of our method:*
>
> Intuitively, our method can be seen as the first pre-training method in the field of certified robustness. The noisy image alignment is considered our *pretext task* in pre-training, while the classification with certified robustness is seen as our *downstream task*. This is analogous to the relationship between contrastive learning methods, such as MoCo-v3, and downstream classification tasks.
>
> Contrastive learning [10, 11, 12] aims at capturing meaningful and general visual representations from natural images, ultimately improving the model’s classification accuracy in downstream image classification tasks. Utilizing infoNCE losses [16], this is achieved by contrasting the semantics of various image pairs in latent space, i.e., each image pair is drawn closer if images within the pair are semantically similar and are repelled otherwise. Similarly, our method computes two infoNCE losses (see also Equation 7), namely, the *consistency* and *contrastive loss*, on image pairs. The image pairs for consistency loss are constructed following the pre-defined noise schedule (see also lines 257-264, “To construct positive pairs”). For the contrastive loss, the image pairs are created utilizing data augmentations, as is common practice in the contrastive learning literature [10, 11, 12].
>
> *Below, we explain the motivation of the key designs of our method:*
>
> - *Why using the PF ODE: lower training difficulty on noisy images.*
>
>     Previous classical approaches (that primarily focus on improving model robustness with different training techniques [8, 13, 14]) are confronted with difficulties when training on noisy images. We argue that this is because they lack principled ways of choosing the noise distributions and training objectives. Therefore, drawing inspiration from the diffusion model literature, we model the relationship between noisy and clean images utilizing the reverse sampling trajectory, where the noisy and clean images are considered as points at different time steps. We pick the PF ODE (i.e., a form of *deterministic* reverse sampling trajectories) to connect each noisy image at a given timestep of the trajectory to a *unique* clean image, which is the solution of the PF ODE at time step 0.
>
> - *Why using consistency losses in latent space: efficiency*
>
>     As discussed in our former response, we use the consistency loss to build semantic connections between clean and noisy images. However, since our final task is classification, denoising an image in pixel space introduces an unnecessary efficiency bottleneck, as demonstrated by diffusion-based methods. As one of our novel contributions, we thus build semantic connections among clean and noisy images in a latent space using an infoNCE loss. This simplifies the task and enables the model to capture meaningful information even from samples with high levels of noise.
>
> - *Why using contrastive loss: preventing training collapse*
>
>     During pre-training, the consistency loss (first term in Equation 7) partitions the sample space by grouping samples according to the PF ODE trajectory they belong to. However, at the start of training, the model tends to output similar representations for different samples. If clean image representations collapse into a single representation in the latent space, training solely on the consistency loss enforces nearly identical representations for all points in the sample space. To address this issue, the contrastive loss (second term in Equation 7) encourages the model to capture meaningful representations from clean images. This helps to effectively differentiate the semantics of initial *clean* points of the PF ODE trajectories, which, due to the consistency loss, also ensures that the model captures more discriminative information on *noisy* samples.
>
> *Conclusion and an empirical view:*
>
> As demonstrated by our experiments, the consistent representations across perturbed and clean samples (acquired during pre-training) significantly lower the training difficulties observed in classical methods [8] when finetuning on noisy samples. Moreover, by learning to denoise in the latent space, we observe robust classification in a single forward pass even for high noise levels, overcoming the costs of diffusion-based methods (relying on two models, iterative denoising, and majority voting). In addition, we empirically observe in our experiments that the performance of our rRCM model has not yet plateaued. In particular, our method can significantly benefit from more training budget, such as model size, training batch size, and total training epochs, providing a principled way of improving model robustness. In summary, our work demonstrates that one could significantly enhance model robustness by leveraging advanced training techniques from both the representation learning and diffusion model literature.

---

> ### Author Response · Authors · 2024-11-29
>
> # Part 4/4
> **Comparison with vanilla ViT**
>
> In Rebuttal-Table-2, the vanilla ViT trained with Gaussian [2] achieves a certified accuracy of 41% at a perturbation radius of 0. As mentioned in Rebuttal-Eq-2, certified accuracies at various perturbation radii are calculated using model evaluations on *noisy* images. In particular, also the certified accuracy at a perturbation radius $r$ of $0$ does *not* represent the classification accuracy on *clean* images but on *noisy* samples at a given noise level $\sigma$.
> Additionally, the certified accuracy is not strongly correlated with the classification accuracy due to the different evaluation procedures of robustness certification and classification, as discussed above (see our explanations for the motivation of our method). Specifically, robustness certification (1) uses noisy inputs, (2) predicts the class with the highest probability among a large number of perturbations, and (3) computes accuracies for different radius thresholds.
>
> We describe the training method in [2] as *heuristic* since it merely adds noise as a data augmentation to an otherwise standard training procedure for classification. While this is a straightforward approach, we have outlined above that this causes training instabilities and struggles to achieve strong robustness (as evidenced by the relatively low certified accuracies in Table 1 of our paper). On the other hand, our work reveals that leveraging the noise schedule of diffusion models for pretraining offers a much more stable, scalable, and theoretically grounded solution.
>
> **Latency**
>
> As a diffusion-based method, the core idea of Carlini et al., 2022 [3] is to purify a perturbed sample using a diffusion model before predicting its class label with a separate classifier. This process incurs at least two forward passes (one for diffusion model $D$, one for classifier $g$) within the base classifier $f$ per prediction. When reproducing [3] and reporting results in Table 1, we follow [3] and use a ViT classifier with the same number of parameters as our rRCM model. However, to obtain strong purification performance, the method in [3] relies on a large diffusion model (552M parameters on ImageNet). This significantly increases the total model size and FLOPS, leading to a longer inference time.
>
> The situation is even more computationally expensive with DensePure [4], where multi-step denoising and majority voting are employed. Specifically, as outlined above, DensePure uses $5$-step denoising combined with majority voting over $K=5$ purified results, resulting in 30 forward passes within the base classifier $f$ per prediction.
>
> In contrast, our method uses a base classifier $f$, which comprises a single, smaller classification model $g$ (~90M parameters) optimized via pre-training and fine-tuning. As a consequence, it can predict the class label of a perturbed sample with just a single model forward pass, substantially reducing inference time.
>
> **Code**
>
> We uploaded our pre-training code to an anonymized repository (link: https://anonymous.4open.science/r/rRCM_for_rebuttal-B603/) to help understand the technical details of our method. We will make our code publicly available upon acceptance. Additionally, for a more detailed understanding of the certification procedure, we refer to the official implementation of randomized smoothing [2], from which we adopted the relevant certification code. The implementation is available at the following link: https://github.com/locuslab/smoothing.
>
> **Additional references** (see the general response for the other references)
>
> [13] SmoothAdv: Provably Robust Deep Learning via Adversarially Trained Smoothed Classifiers: https://arxiv.org/abs/1906.04584
> [14] SmoothMix: Training Confidence-calibrated Smoothed Classifiers for Certified Robustness https://arxiv.org/abs/2111.09277
> [15] MACER: Attack-free and Scalable Robust Training via Maximizing Certified Radius https://arxiv.org/abs/2001.02378
> [16] Representation Learning with Contrastive Predictive Coding https://arxiv.org/abs/1807.03748
>
> ---
>
> Please let us know if you have any further questions or concerns. If our responses have adequately addressed your concerns, we kindly ask you to consider updating your score.

---

> ### Author Response · Authors · 2024-12-02
>
> Dear Reviewer iF9T, As the discussion period is approaching its end, we would like to kindly ask whether we have adequately addressed your concerns. If there are any remaining issues, we would greatly appreciate it if you could share them with us, so we have enough time to provide a comprehensive response.  If our responses have adequately addressed your concerns, we kindly ask you to consider updating your score. Thank you once again for your time and valuable feedback!

---

> > ### Comment · Reviewer_iF9T · 2024-12-02
> >
> > I'd like to thank the authors again for their effort and engagement with my concerns!
> >
> > Your explanations helped me a lot in understanding the motivation of your approach and methodology, now many design choices make more sense to me.
> >
> > I have only the following two remaining comments:
> > - Still, the low performance of the noise-traimed ViT at radius 0.0 is surprising to me. For example, in (https://arxiv.org/abs/1902.02918) fig. 6, we see that all their models have much higher certified accuracy at r=0, and they only use ResNets. However, I understand that training ViTs from scratch needs a lot of data and compute to reach a good performance. In tour comparison, do the ViT and rRCM receive a similar amount of compute? Also, did you consider starting your ViT training from some pre-trained weights to avoid this issue? Also, does your rRCM model start from any pre-trained weights or from scratch?
> > - The code you provided contains only the model definition and training script. Ideally, I'd like to see the "whole thing": training logs, your inference code, experiments, ablation studies etc., in order to further understand some details. For example, related to the above, I wanted to see the code for the ViT vs rRCM and check some of these details. Would it be possible to update the link and add the rest? No need to be clean!
> >
> > These are my only remaining concerns, thank you very much again!

---

> ### Author Response · Authors · 2024-12-02
>
> We sincerely thank you for your insightful feedback! Below, we provide additional experimental details and clarifications for our comparisons with the ViT baseline trained using Gaussian [2]. We are currently collecting and anonymizing the logs and the remaining parts of our code and will upload them as soon as possible. **However, since the reviewer response period is about to end, we wanted to post the response as soon as possible to clarify any remaining questions. If required, we are happy to provide more detailed explanations until the final author response deadline.**
>
> **Experiment settings for comparing with ViT baseline:**
> The ViT baseline reported in Rebuttal-Table-2 was trained from scratch for 400 epochs with a batch size of 1024, using the official implementation of Gaussian [2] (link: https://github.com/locuslab/smoothing). Noticeably, the ViT baseline reached its plateau very quickly, and longer training did not lead to further improvement in certified robustness. In comparison, our rRCM-B model was fine-tuned for 150 epochs with the same batch size.
>
> We agree with your statement that training ViTs from scratch demands an extensive amount of data. Moreover, we believe the lower certified accuracy at radius = 0.0 can also be attributed to the following two factors, which are amplified for ViTs by the lack of a strong architectural inductive bias (compared to ResNets):
> (1) The inherent difficulty of directly training models on noisy images (as outlined in our previous response).
> (2) The absence of tailored data augmentations in the official Gaussian [2] implementation, which ViT typically requires.
>
> **Training ViT baseline with pre-trained weights:** Our primary goal has been to illustrate the challenges of training models from scratch on noisy images using the Gaussian [2] method. Hence, we did not use pre-trained weights for the ViT baseline in these experiments. However, in our paper, we provide comparisons with MoCo-v3, additionally equipped with Gaussian noise as a baseline. Specifically, we pre-trained a ViT model (which has the same number of parameters as rRCM-B) using MoCo-v3 with Gaussian noise as additional augmentation, then fine-tuned it under the same settings as our rRCM-B model. In particular, we use the same compute budget for pre-training and fine-tuning MoCo-v3 and our rRCM-B. As shown in Figure 9 of our paper, our rRCM-B model significantly outperforms this baseline, providing relevant insights into the advantages of our proposed pre-training approach. **Please let us know if this addresses your concerns regarding a comparison to ViT models with pre-trained weights.**
>
> **Whether our rRCM model needs pre-trained weights:** Our rRCM models are trained entirely from scratch during the pre-training phase (i.e., without using pre-trained weights). Subsequently, the rRCM models, reported in Table 1, Table 2,  Rebuttal-Table-1, and Rebuttal-Table-2, are fine-tuned starting from the weights obtained from this pre-training phase.

---

> > ### Comment · Reviewer_iF9T · 2024-12-03
> >
> > Would it be possible to do the same experiment with a pre-trained ViT and see what the results will be? Or, if others have already tried this, you can also point out their papers to take a look. I think this is an important comparison. Also, would rRCM benefit if started from some pre-trained weights?
> >
> > Apart form that no other concerns! I'll take a look at the updated paper, code and the comments here and revise my review. Thanks a lot again for your effort!

---

> ### Author Response · Authors · 2024-12-04
>
> We sincerely thank you for your time and constructive feedback!
>
> **Comparison with Gaussian [2] using a ViT-B model with pre-trained weights:** As suggested by you, we have run a new experiment for the Gaussian method [2] with a ViT model (having the same number of parameters as our rRCM-B) that is loaded with weights pre-trained on ImageNet-21k (Link to the weights: https://github.com/lukemelas/PyTorch-Pretrained-ViT).
> The model is subsequently fine-tuned and certified separately at noise levels $\sigma \in \\{0.25, 1.0\\}$ for 150 epochs with a batch size of 1024 (same settings as our rRCM-B model) on ImageNet-1k. We present the certified accuracy of these two models in Rebuttal-Table-2.  For your convenience, the table is also included below.
>
> Rebuttal-Table 2: Comparison with Gaussian [2] ($\sigma=0.25$):
>
> Model Name  &nbsp;&nbsp;&nbsp;&nbsp;&nbsp;&nbsp;&nbsp;&nbsp;&nbsp;&nbsp;&nbsp;&nbsp;&nbsp;&nbsp;&nbsp;&nbsp;&nbsp;&nbsp;&nbsp;&nbsp;&nbsp;&nbsp;&nbsp;&nbsp;&nbsp;&nbsp;&nbsp;&nbsp;&nbsp;&nbsp;&nbsp;&nbsp;&nbsp;&nbsp;&nbsp;&nbsp;&nbsp;&nbsp;&nbsp;&nbsp;&nbsp;&nbsp;&nbsp;&nbsp;&nbsp;&nbsp;&nbsp;&nbsp;&nbsp; Certified accuracy at r (%)
> |  | |  |  |   |   |   |
> | --- | --- | --- | --- | --- | --- |  --- |
> |  | 0.0 | 0.5 |
> | Gaussian [2] | 41.0 | 27.0 |
> | Gaussian [2] - pretrained on ImageNet-22k| 74.5 | 52.1 |
> | Ours: rRCM-B| 76.6 | 62.6 |
>
>
> Comparison with Gaussian [2] ($\sigma=1$):
>
> Model Name &nbsp;&nbsp;&nbsp;&nbsp;&nbsp; &nbsp;&nbsp;&nbsp;&nbsp; &nbsp;&nbsp;&nbsp;&nbsp;&nbsp;&nbsp;&nbsp;&nbsp;&nbsp;&nbsp;&nbsp;&nbsp;&nbsp;&nbsp;&nbsp;&nbsp;&nbsp;&nbsp;&nbsp;&nbsp;&nbsp;&nbsp;&nbsp;&nbsp;&nbsp;&nbsp;&nbsp;&nbsp;&nbsp;&nbsp;&nbsp;&nbsp;&nbsp;&nbsp;&nbsp;&nbsp;&nbsp;&nbsp;&nbsp;&nbsp;&nbsp;&nbsp;&nbsp;&nbsp;&nbsp;&nbsp;&nbsp;&nbsp; Certified accuracy at r (%)
> |  | |  |  |   |   |   |   |
> | --- | --- | --- | --- | --- | --- |  --- | --- |
> |  | 0.0 | 0.5 | 1.0 | 1.5 | 2.0 | 2.5 |
> | Gaussian [2] - pretrained on ImageNet-22k| 10.6 | 6.8 | 5.6 | 2.4 | 1.6 | 1.0 |
> | Ours: rRCM-B| 52.6 | 45.6 | 39.4 | 33.8 | 27.0 | 22.0 |
>
> **Notice: Due to the short amount of time available, we certify the models using 10k smoothing noises. As a result, the outcomes are not directly comparable to those reported in Gaussian [2], where they use 100k smoothing noises during the certification process.**
>
> As demonstrated, at the smaller noise level $\sigma=0.25$, the ViT-B significant benefits from pre-training on a large-scale dataset, reaching a certified robustness of 74.5% at perturbation radius=0. However, at a higher noise level of $\sigma = 1.0$, the model exhibits extremely low certified accuracy. Our observations indicate that during fine-tuning at $\sigma = 1.0$, the training loss decreases very slowly and quickly plateaus across all tested hyperparameter settings.
>
> We speculate that this is primarily due to the gap between pre-training on clean (or augmented) images and fine-tuning on noisy images. During pre-training, the model learns representations based on meaningful semantics in the clean input images. However, during fine-tuning, strong noise is added to the images, severely corrupting these semantics. At lower noise magnitudes, where the images still retain distinct semantic features, the model can leverage pre-trained weights effectively. Conversely, at higher noise levels, the corrupted semantics significantly diminish the advantages of pre-training.
>
> In contrast, our rRCM pre-training is explicitly designed to leverage multiple (including very high) noise levels, leading to smooth convergence on noisy images during fine-tuning and achieving superior certified accuracy.
>
> It is worth noting that the pre-trained weights used for fine-tuning the ViT-B model are derived from training on ImageNet-21k, while our rRCM-B model is pre-trained on the smaller ImageNet-1k dataset. ImageNet-21k contains 21,000 classes and 14 million images, making it approximately 10 times larger than ImageNet-1k, which includes only 1,000 classes and 1.3 million images. **We will run further ablations with other pre-trained weights and more noise levels for the final version**.
>
>
> **Code and logs:** We uploaded our codes and logs to the anonymized repository (link: https://anonymous.4open.science/r/rRCM_for_rebuttal-B603). For readability, we provide a readme.md file to explain the structure of files. In particular, we have uploaded the training logs of our rRCM-B model and our codes used in pre-training, fine-tuning, and certification. Besides, we have also uploaded codes for re-producing the ViT baseline experiment in Rebuttal-Table-2, along with the training logs. We have also uploaded the code and logs for the new experiment with pretrained weights, which is based on the official implementation of Gaussian [2] (link: https://github.com/locuslab/smoothing).

---

> ### Author Response · Authors · 2024-12-04
>
> **Train rRCM with pre-trained weights**  In our initial experiments, we observed significant training challenges, such as NaN loss, when attempting to pre-train the rRCM model using weights derived from other pre-training tasks. We speculate that this issue arises from the discrepancy between the objectives of our pre-training task and those of the pre-trained weights.

---

### Author Response · Authors · 2024-11-23
**General Response to our Reviewers (Part 1/2)**

We thank the reviewers for their valuable comments, which helped to significantly improve our paper. In this response, we present additional experimental results to further demonstrate the advantages of our proposed method. Besides, we answer common questions and summarize our modifications of the paper. Finally, we list the references that have been cited in our responses.

## Part 1/2
**Experiments**

- Generalizability to another dataset

We conduct additional experiments on the medical image dataset *KatherColon* [1], which consists of 10,000 224x224 images categorized into 9 classes. We fine-tune the rRCM-B model pre-trained on ImageNet to evaluate the generalizability of the pre-trained model weights. For comparison, we include results from DDS, citing the reported numbers from [1]. During certification, we use 10,000 smoothing noises.

**Rebuttal-Table 1**: Comparing with DDS [3] on medical image dataset KatherColon.

Model Name &nbsp;&nbsp;&nbsp;&nbsp;&nbsp;&nbsp;&nbsp;&nbsp;&nbsp;&nbsp;&nbsp;&nbsp;&nbsp;&nbsp;&nbsp;&nbsp;&nbsp;&nbsp;&nbsp;&nbsp; Certified accuracy at r (%)
|  | 0.0 | 0.25 | 0.5 | 0.75 | 1.0 | 1.25 | 1.5 |
| --- | --- | --- | --- | --- | --- |  --- | --- |
| DDS | 58.0 | 49.0 | 41.0 | 34.0 | 26.0 | 22.0 | 16.0 |
| Ours: rRCM-B | 87.4 | 80.6 | 72.4 | 59.4 | 50.2 | 48.2 | 45.4 |

- Comparing with classical methods using ViT architecture
We re-implement and compare with Gaussian [2] using the ViT-B architecture, which has the same number of parameters as our rRCM-B model. In the limited time of the rebuttal, we train and certify models separately at noise level $\sigma=0.25$ and $\sigma=1.0$. However, we will add experiments at other noise levels (e.g., 0.5) and with more pre-trained weights in the final version. We report the certified accuracy under different perturbation radii utilizing 10,000 smoothing noises.

Rebuttal-Table 2: Comparison with Gaussian [2] ($\sigma=0.25$):

Model Name &nbsp;&nbsp;&nbsp;&nbsp; &nbsp;&nbsp;&nbsp;&nbsp; &nbsp;&nbsp;&nbsp;&nbsp;&nbsp;&nbsp;&nbsp;&nbsp;&nbsp;&nbsp;&nbsp;&nbsp;&nbsp;&nbsp;&nbsp;&nbsp;&nbsp;&nbsp;&nbsp;&nbsp;&nbsp;&nbsp;&nbsp;&nbsp;&nbsp;&nbsp;&nbsp;&nbsp;&nbsp;&nbsp;&nbsp;&nbsp;&nbsp;&nbsp;&nbsp;&nbsp;&nbsp;&nbsp;&nbsp;&nbsp;&nbsp;&nbsp;&nbsp;&nbsp;&nbsp;&nbsp;&nbsp;&nbsp;&nbsp; Certified accuracy at r (%)
|  | |  |  |   |   |   |
| --- | --- | --- | --- | --- | --- |  --- |
|  | 0.0 | 0.5 |
| Gaussian [2] | 41.0 | 27.0 |
| Gaussian [2] - pretrained on ImageNet-22k| 74.5 | 52.1 |
| Ours: rRCM-B| 76.6 | 62.6 |


Comparison with Gaussian [2] ($\sigma=1$):

Model Name &nbsp;&nbsp;&nbsp;&nbsp;&nbsp; &nbsp;&nbsp;&nbsp;&nbsp; &nbsp;&nbsp;&nbsp;&nbsp;&nbsp;&nbsp;&nbsp;&nbsp;&nbsp;&nbsp;&nbsp;&nbsp;&nbsp;&nbsp;&nbsp;&nbsp;&nbsp;&nbsp;&nbsp;&nbsp;&nbsp;&nbsp;&nbsp;&nbsp;&nbsp;&nbsp;&nbsp;&nbsp;&nbsp;&nbsp;&nbsp;&nbsp;&nbsp;&nbsp;&nbsp;&nbsp;&nbsp;&nbsp;&nbsp;&nbsp;&nbsp;&nbsp;&nbsp;&nbsp;&nbsp;&nbsp;&nbsp;&nbsp; Certified accuracy at r (%)
|  | |  |  |   |   |   |   |
| --- | --- | --- | --- | --- | --- |  --- | --- |
|  | 0.0 | 0.5 | 1.0 | 1.5 | 2.0 | 2.5 |
| Gaussian [2] - pretrained on ImageNet-22k| 10.6 | 6.8 | 5.6 | 2.4 | 1.6 | 1.0 |
| Ours: rRCM-B| 52.6 | 45.6 | 39.4 | 33.8 | 27.0 | 22.0 |

**Questions**

- Why generation experiment:

We conduct the image generation experiment to showcase that our method learns consistent representations among perturbed samples and clean images. This demonstrates that our performance gains are not simply brought by pre-training on noisy images and verifies that our optimization objective does achieve the anticipated results. That is, the representations of any temporally adjacent points on the same PF ODE trajectory are strongly aligned, such that $f(x_t)\cdot f(x_{t-1}) \approx 1$. The image generation experiment also demonstrates that the consistency is strong enough such that our model can generate clean representations given Gaussian noise as input (see also Appendix D and Table 6).

In detail, we train a diffusion model conditioned on the output of our rRCM-B model, which takes clean images as input (as in [7]). After that, we generate images using the trained diffusion model conditioned on representations obtained by rRCM-B from pure Gaussian noise. We observe that the quality of the images generated by the diffusion model does not degrade substantially, showing that these representations do not differ significantly from those extracted from clean images.

Beyond the scope of certified robustness, our method serves as a preliminary exploration of training a model capable of learning both strong representations and the corresponding distribution in the latent space in a way that is analogous to denoising training in the pixel space.

---

> ### Author Response · Authors · 2024-11-23
> **General Response to our Reviewers (Part 2/2)**
>
> ## Part 2/2
>
> - Configuration we use during inference
> We measure the inference time of each method using a single A800 GPU (80GB memory). During certification, the batch sizes are chosen to ensure an optimal inference time is achieved for each baseline method. Specifically, for DDS [3], DiffSmooth [4], DensePure [5], and our models, we use a batch size of 100 on ImageNet, and 500 on CIFAR10. Besides, we evaluate the inference time of Gaussian [2] (26M ResNet50 on ImageNet, 46M RestNet101 on CIFAR10) with a batch size of 4000 on ImageNet and 1000 on CIFAR10. We re-use the results measured on Gaussian for other classical methods while, in particular, accounting for the model ensembling strategy used in [6]. This is because, during evaluation, all classical methods (except [6]) listed in Table 1 utilize a single model (ResNet50 on ImageNet and ResNet101 on CIFAR10) and predict class labels within a single forward pass. As a result, the magnitude of their theoretical inference latency is similar (and our primary focus was to compare our latency to state-of-the-art diffusion-based methods).
>
> - z1 and z2 in pseudocode
> z1 and z2 are two augmented views of the same clean image x0. We have updated our pseudocode along with the presentation of our pre-training method in Section 3.3, making it more accessible.
>
> **Modification to the paper**
> - We add explanations of why our method is preferable compared to the classical methods in lines 87-90 and lines 111-122.
> - We introduce details of our fine-tuning method along with the underlying rationale in Section 3.4
> - We made significant revisions to the problem formulation in Section 3.2 and the pre-training methodology in Section 3.3.
> - We defer additional experiments, including *image generation* and *comparison to MoCo-v3 [11]*, to our appendix since they are not the primary focus of our work.
>
> **References**
> [1] PromptSmooth: Certifying Robustness of Medical Vision-Language Models via Prompt Learning: https://www.arxiv.org/abs/2408.16769
> [2] Certified Adversarial Robustness via Randomized Smoothing: https://arxiv.org/abs/1902.02918
> [3] (Certified!!) Adversarial Robustness For Free!: https://arxiv.org/abs/2206.10550
> [4] DensePure: Understanding Diffusion Models Towards Adversarial Robustness:https://arxiv.org/abs/2211.00322
> [5] DiffSmooth: Certifiably Robust Learning via Diffusion Models and Local Smoothing:https://www.usenix.org/conference/usenixsecurity23/presentation/zhang-jiawei
> [6] Boosting randomized smoothing with variance reduced classifiers:https://arxiv.org/abs/2106.06946
> [7] Return of Unconditional Generation: A Self-supervised Representation Generation Method:https://arxiv.org/abs/2312.03701
> [8] Consistency Regularization for Certified Robustness of Smoothed Classifiers: https://arxiv.org/abs/2006.04062
> [9] Github code repository of Consistency Model: *https://github.com/openai/consistency_models*
> [10] A Simple Framework for Contrastive Learning of Visual Representations:https://arxiv.org/abs/2002.05709
> [11] An Empirical Study of Training Self-Supervised Vision Transformers:https://arxiv.org/abs/2104.02057
> [12] Exploring Simple Siamese Representation Learning:https://arxiv.org/abs/2011.10566.

---

### Meta-Review · Area_Chair_uyya · 2024-12-20

**Metareview:**

This paper introduces the Robust Representation Consistency Model (rRCM), a method that combines ideas from contrastive learning and diffusion models to improve certified adversarial robustness in deep neural networks. Instead of relying on traditional generative denoising, the authors reframe the task as a discriminative problem in latent space, which allows the model to denoise and classify in a single forward pass. This approach is a concrete step forward, offering both strong robustness across a wide range of perturbation radii and significantly reduced inference costs. The experiments on large-scale datasets like ImageNet and CIFAR10 provide compelling evidence that rRCM achieves state-of-the-art certified accuracy while being much more computationally efficient.

The paper has some clear strengths. The proposed method addresses a significant gap between the efficiency of classical approaches and the robustness of diffusion-based methods. The idea of leveraging PF-ODE trajectories and consistency learning in latent space is novel and well-motivated, and the empirical results back this up with consistent improvements across the board. The authors also went above and beyond during the review process, conducting additional experiments, clarifying confusing parts of the methodology, and even providing anonymized code and logs to address concerns about implementation and baseline comparisons.

That said, there are a few areas where the paper could be stronger. Initially, the presentation of the methodology was a bit difficult to follow, especially for reviewers less familiar with the field. Some reviewers also raised concerns about whether baseline models like Gaussian and MoCo-v3 were being fairly compared. The authors tackled these issues head-on by revising the methodology section, updating pseudocode and figures, and running additional experiments.

In the end, the strengths of this paper outweigh the weaknesses. It brings a novel, scalable, and effective solution to a challenging problem, and the authors’ thorough responses during the review process further solidify its contribution. This is a strong paper, and I recommend accepting it.

**Additional Comments On Reviewer Discussion:**

The review process raised several critical points that shaped the final decision. Reviewer iF9T initially struggled to understand the motivation for the method and asked why simpler baselines like training on noisy images weren’t enough. The authors provided additional experiments and theoretical explanations, making it clear why their approach is superior. Reviewer Svp1 had questions about the fairness of the MoCo-v3 comparison and the necessity of the image generation experiment. The authors clarified these points by adding experimental details and moving the less-relevant parts, like image generation, to the appendix. Reviewer wTyH flagged issues with the clarity of the methodology, which led to significant improvements in the writing and the inclusion of clearer pseudocode and figures.

The authors handled the discussion phase well, addressing all major concerns with care and providing detailed, thoughtful responses. Reviewers acknowledged the improvements, with some even raising their scores. While there were lingering concerns about specific details, the overall consensus was that this paper represents a strong and meaningful contribution. Based on the discussion and revisions, I confidently recommend accepting this work.

---

### Decision · Program_Chairs · 2025-01-22

Accept (Poster)